# Leaky Gut Biomarkers as Predictors of Depression and Suicidal Risk: A Systematic Review and Meta-Analysis

**DOI:** 10.3390/diagnostics15131683

**Published:** 2025-07-01

**Authors:** Donato Morena, Matteo Lippi, Matteo Scopetti, Emanuela Turillazzi, Vittorio Fineschi

**Affiliations:** 1Department of Anatomical, Histological, Forensic and Orthopedic Sciences, Sapienza University of Rome, 00161 Rome, Italy; matteo.lippi@uniroma1.it (M.L.); vittorio.fineschi@uniroma1.it (V.F.); 2Department of Medical Surgical Sciences and Translational Medicine, Sapienza University of Rome, 00189 Rome, Italy; matteo.scopetti@uniroma1.it; 3Department of Surgical Pathology, Medical, Molecular and Critical Area, University of Pisa, 56126 Pisa, Italy; emanuela.turillazzi@unipi.it

**Keywords:** gut–brain axis, dysbiosis, intestinal permeability, depressive symptoms, suicidality, biomarkers, inflammation, forensic medicine

## Abstract

**Background**: The gut–brain axis (GBA) has been demonstrated to be involved in normal neurodevelopment, with its dysfunction potentially contributing to the onset of mental disorders. In this systematic review and meta-analysis, we aimed to examine the relationship between levels of specific biomarkers of intestinal permeability or inflammation and scores of depressive symptoms or suicidality. **Methods**: All studies investigating the link between depressive symptoms and/or suicidality and biomarkers associated with intestinal permeability or inflammation were included. Studies providing data for comparisons between two groups—depressive or suicidal patients vs. healthy controls, or suicidal vs. non-suicidal patients—were included in the meta-analysis. Studies examining the correlation between depressive symptoms and biomarker levels were also included into the review. Data were independently extracted and reviewed by multiple observers. A random-effects model was employed for the analysis, and Hedge’s g was pooled for the effect size. Heterogeneity was assessed using the I^2^ index. **Results**: Twenty-two studies provided data for inclusion in the meta-analysis, while nineteen studies investigated the correlation between depressive symptoms and biomarker levels. For depressive symptoms, when compared to the controls, patients showed significantly increased levels of intestinal fatty acid-binding protein (I-FABP) (ES = 0.36; 95% CI = 0.11 to 0.61; *p* = 0.004; I^2^ = 71.61%), zonulin (ES = 0.69; 95% CI = 0.02 to 1.36; *p* = 0.044; I^2^ = 92.12%), antibodies against bacterial endotoxins (ES = 0.75; 95% CI = 0.54 to 0.98; *p* < 0.001; I^2^ = 0.00%), and sCD14 (ES = 0.11; 95% CI = 0.01 to 0.21; *p* = 0.038; I^2^ = 10.28%). No significant differences were found between the patients and controls in levels of LPS-binding protein (LBP) and alpha-1 antitrypsin (A-1-AT). For suicidality, four studies were identified for quantitative analysis, three of which focused on I-FABP. No significant differences in I-FABP levels were observed between suicidal patients and the controls (ES = 0.24; 95% CI = −0.30 to 0.79; *p* = 0.378; I^2^ = 86.44%). Studies investigating the correlation between depressive symptoms and levels of intestinal permeability and inflammation biomarkers did not provide conclusive results. **Conclusions**: A significant difference was observed between patients with depressive symptoms and controls for biomarkers of intestinal permeability (zonulin, which regulates tight junctions), inflammatory response to bacterial endotoxins (antibodies to endotoxins and sCD14—a soluble form of the CD14 protein that modulates inflammation triggered by lipopolysaccharides), and acute intestinal epithelial damage (I-FABP, released upon enterocyte injury). Studies investigating suicidality and related biomarkers were limited in number and scope, preventing definitive conclusions. Overall, these findings suggest that biomarkers of gut permeability represent a promising area for further investigation in both psychiatric and forensic pathology. They may have practical applications, such as supporting diagnostic and therapeutic decision-making in clinical settings and providing pathologists with additional information to help determine the manner of death in forensic investigations.

## 1. Background

Depression is one of the most prevalent mental disorders, affecting approximately 280 million people worldwide [1] and contributing to 56.3 million years lived with disability [2]. Its pathogenic mechanisms are increasingly recognized as complex and multifactorial [3].

Traditionally, depression has been linked to dysfunctions in neurotransmitter systems, particularly a deficiency in monoamines such as serotonin [4] and noradrenaline [5]. Subsequent research has expanded this framework to include dopaminergic [6] and glutamatergic transmission [7]. These foundational hypotheses of neurotransmitter dysfunction have been further refined by findings highlighting additional mechanisms, including impaired neuroplasticity [8] and immunomodulation [9,10]. A growing body of evidence supports the interconnection between the immune system and various mental disorders [11,12,13], alongside complex interactions between immune function, the gut, and the microbiome [14]. Neuroinflammation, involving systems such as the hypothalamic–pituitary–adrenal (HPA) axis and the gut–brain axis (GBA), has emerged as a potential neurobiological correlate of depression [15,16]. Structural changes within these systems, along with dysregulation of bidirectional signaling, are believed to impair brain function [17,18].

Regarding the HPA axis, chronic hyperactivity has been linked to glucocorticoid resistance, a condition in which glucocorticoid hormones fail to exert their usual effects on target tissues, potentially leading to immune activation. This immune activation can, in turn, further amplify inflammation, creating a feedback loop that exacerbates HPA axis dysfunction through the direct action of cytokines on brain regions and the induction of glucocorticoid resistance [9,19].

The HPA axis also responds to gut-derived signals, modulating cortisol secretion, which in turn influences gut motility, permeability, and immune responses. The GBA functions as a bidirectional communication network that integrates neural, endocrine, immune, and metabolic pathways, all of which contribute to the regulation of homeostasis and brain function [20].

The microbiota serves as an intermediary within this axis, supplying neuroactive and immunomodulatory substances [21]. It plays a crucial role in maintaining intestinal barrier integrity [22] and has been linked to the development and function of the nervous system through its interaction with the GBA [23]. This interaction is bidirectional, with changes in microbiota composition influencing behavior, while behavioral modifications induced by chronic stress, genetic variation, or pharmacological intervention can, in turn, alter the microbiota [24,25]. Due to the complexity of these interactions, it has been proposed to expand the GBA model into a “microbiome–GBA” framework [23].

Intestinal homeostasis—referring to the healthy, balanced state of the gut—is maintained by the intestinal epithelium, the gut microbiome, and the host immune system. Under physiological conditions, the intestinal barrier maintains a balance between absorbing dietary nutrients into the systemic circulation and protecting the body from pathogens and harmful external components. Disruption of intestinal barrier integrity and dysfunction can permit the uncontrolled passage of bacterial components, metabolic products, and harmful substances, leading to systemic inflammation. Alterations in gut microbiota composition, known as dysbiosis, can compromise intestinal barrier integrity and promote the overgrowth of Gram-negative lipopolysaccharide (LPS)-producing Proteobacteria. This leads to an increase in endotoxins that translocate into the systemic circulation [26,27]. LPS and other endotoxins play a key role in the initiation and progression of low-grade systemic inflammation [28], triggering the release of pro-inflammatory cytokines that adversely affect the brain through neuroinflammatory mechanisms [29].

Recent evidence has further clarified the bidirectional interactions within the microbiome–GBA, highlighting intestinal permeability [30]—commonly referred to as the “leaky gut” hypothesis—as a key mechanism contributing to systemic and neuroinflammatory processes [31].

This condition allows bacteria, toxins (e.g., LPS), and small molecules to translocate into the bloodstream. The resulting systemic inflammation triggers the release of pro-inflammatory cytokines (e.g., IL-6 and IL-1β), which can cross or disrupt the blood–brain barrier (BBB), leading to neuroinflammation [32]. Animal studies suggest that dysbiosis is associated with increased permeability of the BBB [33], indicating a functional link between the two barriers. Furthermore, intestinal barrier dysfunction can also impact the regulation of neurotransmitter systems, such as neuroamines, which are crucial for mood regulation [34].

Recent studies using animal models have demonstrated that in the presence of intestinal inflammation and compromised local vascular protection—specifically, dysfunction of the gut–vascular barrier [35]—there are also distant alterations in the vascular barrier of the choroid plexus. This barrier plays a critical role in shielding the brain from inflammatory insults, and its dysfunction may potentially contribute to impaired brain function [36].

Beyond efforts to integrate emerging pathogenic hypotheses, there is growing scientific interest in identifying reliable biomarkers for mental disorders [37], particularly depressive disorders [38,39], with a specific focus on depression and suicide risk [40,41]. Most mental disorders, particularly severe ones [42], are associated with an increased risk of suicide. Compared to the general population, individuals with major depressive disorder (MDD) have a twenty-fold increased risk of suicide [43]. Suicide is a significant public health issue worldwide, affecting individuals of all ages, and it is one of the leading causes of death among younger populations [44].

Stratifying risk remains valuable, and biomarkers have the potential to enhance diagnostic precision, predict disease progression, and inform personalized therapeutic strategies.

In this evolving context, the present study aimed to investigate the potential role of biomarkers related to intestinal permeability and systemic inflammation as correlates of depressive symptoms and suicidality. Unlike previous research that has primarily focused on diagnostic categories [45], in our study a dimensional approach was adopted, examining the depressive spectrum and suicidality as continuous, clinically relevant constructs. The depressive dimension is increasingly conceptualized as a continuum of symptom severity, rather than a discrete diagnostic category. This perspective encompasses subclinical states, major depressive episodes, and chronic forms of depression, each associated with distinct symptom profiles and variable impacts on function and prognosis. This dimensional approach is supported by empirical evidence from epidemiological, clinical, and neurobiological studies, which suggest overlapping mechanisms across the spectrum of depressive disorders [46,47].

In summary, the central hypotheses of this study were as follows:i)Biomarkers indicative of increased intestinal permeability and bacterial translocation are positively associated with the severity of depressive symptoms;ii)These biomarkers are also elevated in individuals with a history or risk of suicidality, independently of diagnostic classification.

A summary of the primary biomarkers examined is presented in Table 1 (with a complete description of all biomarkers investigated in the systematic review in Appendix A).

To the best of our knowledge, no systematic evidence on these trans-nosographic aspects is currently available in the literature.

Furthermore, this investigation may have significant implications in forensic pathology, where biomarkers could serve as valuable tools for determining the manner of death and profiling individuals at risk of suicide.

## 2. Methods

This systematic review and meta-analysis was conducted following the PRISMA guidelines (the PRISMA checklist can be consulted in Appendix A). A search for published studies was conducted from inception to July 2024 in PubMed, Scopus, Embase, the Cochrane Register of Controlled Trials, and Google Scholar. Two independent raters (D.M. and M.L.) extracted relevant study characteristics and outcome measures data. Disagreements during both screening and data extraction were resolved through discussion, with arbitration by senior review authors (E.T. and V.F.), considering the inclusion criteria (Appendix A) and the relevance of the data to our study. Emerging reviews and the reference lists of the retrieved papers were also manually searched by two investigators (D.M. and M.L.).

Studies were included if they i) adopted a case–control design and reported data on easily accessible biomarkers (e.g., blood, serum, plasma, or fecal) related to intestinal permeability and bacterial translocation and ii) involved participants diagnosed with mood disorders according to internationally validated criteria, with current depressive symptoms. Studies involving patients with a recent suicide attempt, current suicidal ideation, or elevated suicidality—assessed via standardized tools—were also considered (for a detailed description of the inclusion and exclusion criteria, see Appendix A).

Both cross-sectional studies and studies with various protocols were analyzed, with baseline data extracted for the latter. Studies providing data for comparisons between two groups—depressive or suicidal patients vs. healthy controls (HCs), or suicidal vs. non-suicidal patients—were included in the meta-analysis. Studies assessing the correlation between depressive symptoms and biomarker levels were also included in the review. Our search string included all intestinal permeability or inflammation biomarkers related to bacterial translocation known in the literature to the best of our knowledge (Appendix A).

To ensure greater homogeneity in the laboratory analysis methods, a time filter was applied to exclude articles published before January 2000. Papers not written in English or not published in peer-reviewed journals were also excluded.

For the full dataset of studies, see Appendix A.

The protocol for this review has been registered with the International Prospective Register of Systematic Reviews (PROSPERO registration number CRD42024626391).

All authors of studies from which unpublished data could be obtained were contacted via email twice. When relevant information required for inclusion in our study could not be obtained from the authors, a meticulous review of the studies was conducted through a conference between the raters (D.M. and M.L.) and the senior authors (E.T. and V.F.).

The PRISMA flow chart of the study can be consulted in Figure 1.

## 3. Quality of Evidence

The quality assessment of the studies included in the quantitative analysis was conducted using the Critical Appraisal Skills Programme (CASP, Oxford, UK) checklist for cross-sectional studies [48,49]. Despite the heterogeneity in study designs (14 cross-sectional studies, 5 case–control studies, 2 cohort studies, and 1 randomized controlled trial), we opted to extract data exclusively at the baseline measurement to ensure consistency across the datasets. Consequently, we decided to use only the CASP checklist for cross-sectional studies to maintain uniformity in the risk of bias assessment.

## 4. Outcomes

Our primary outcome was to compare differences in circulating levels of proxy biomarkers between depressive or suicidal patients vs. HCs or between suicidal vs. non-suicidal patients. Our secondary outcomes aimed to highlight the degree and significance of the association between the severity of depressive symptoms and/or suicidality and the levels of proxy biomarkers. A quantitative synthesis of the differences in circulating levels was performed when data from three or more studies were available for a specific biomarker. Medians, interquartile ranges (IQRs), and standard errors (SEs) were converted to means and standard deviations (SDs) using validated methods to standardize data for meta-analysis [50]. Conversions followed established formulas (e.g., SD = SE × √n) and assumed approximate data symmetry. While estimations may introduce minor uncertainty, they are widely accepted to enhance comparability and data inclusion across studies.

When necessary, data were extracted from graphs using a web-based tool (WebPlot Digitizer v5.2^®^).

In our procedure, we first computed effect sizes (ESs) for each study to address our research questions. Secondly, we combined the ESs across studies using the inverse-variance method [51]. The random-effects model was used as a conservative approach to account for various sources of variation among studies (i.e., within-study and between-study variance) and to allow for generalization of the meta-analytic findings beyond the studies included in the current review [52]. Values of 0.20, 0.50, and 0.80 for Hedges’ g were considered indicative of small, medium, and large effects, respectively [53]. In contrast to other similar meta-analytic studies, we used Hedges’ g instead of the Standardized Mean Difference (SMD), as it provides a more accurate estimate of the ES in the presence of small sample sizes by correcting for bias in the SD [51,54].

Sensitivity analyses were performed by sequentially removing individual studies and re-running the analysis. Moderator analyses were conducted to assess whether the severity of depressive symptoms or suicidality could identify distinct outcome patterns among participants and provide a basis for determining the populations in which the use of biomarkers is most reliable [55]. For this purpose, we categorized the scores for depressive symptoms and suicidal risk into five discrete subgroups of severity, based on the categories of the individual scales used in the studies. The moderator analysis was conducted when at least three studies were available for each level of the moderator [55]. Heterogeneity among studies in each meta-analysis was assessed using the chi-squared statistic (Q), I^2^, and Tau^2^. Substantial heterogeneity was considered if I^2^ exceeded 30%, and either Tau^2^ was greater than zero or there was a low *p*-value (less than 0.10) in the chi-squared statistic (Q) test for heterogeneity. I^2^ represents the proportion of observed variability that is real rather than spurious, indicating the extent to which the observed variance is due to true effect variation rather than sampling error [51]. It is scale-independent and not influenced by low statistical power. We considered I^2^ values as low when ranging from 0 to 25%, intermediate from 25 to 50%, moderate from 50 to 75%, and high when ≥75% [55]. Finally, the risk of publication bias was evaluated through i) a visual examination of funnel plots, ii) a statistical test of asymmetry (Egger’s test) [56], and iii) the trim and fill procedure [57]. For all analyses, a *p*-value < 0.05 (two-tailed) was considered statistically significant. All analyses were conducted using the meta-analytic software ProMeta3^®^.

## 5. Results

The searches identified 1958 records. After removing duplicates and examining titles and abstracts, we selected 123 records for full-text assessment (Appendix A). Ultimately, 26 studies that compared patients and controls were selected (Table 2), of which 22 provided data for inclusion in the meta-analysis (21 focused on depressive symptoms and 4 on suicidality). For depressive symptoms, a quantitative synthesis was possible for six biomarkers: intestinal fatty acid-binding protein (I-FABP), zonulin, LPS-binding protein (LBP), antibodies against bacterial endotoxins, alpha-1 antitrypsin (A-1-AT), and soluble CD14 (sCD14) (Figure 2). 

For suicidal risk, a quantitative synthesis was possible only for I-FABP (Figure 3).

Nineteen studies were also identified that assessed the correlation between depressive symptoms and proxy biomarker levels (Table 3).

The results of the quality of evidence are summarized in Appendix A: all studies achieved a score on the CASP checklist of 8 or higher, indicative of high quality, except for two studies (Maes et al. 2008 [58], Maes et al. 2019 [59]), which scored 7 (medium quality). Further quantitative analyses are provided in the Appendix A. Publication bias analyses for primary outcomes are reported in Appendix A. 

In Table 2 and Table 3, a comprehensive summary of the findings from all retrieved studies is provided. The summary includes data on the type of biomarker and biological sample analyzed, patients’ sociodemographic characteristics, type of mental disorder, stage of illness, assessment tools employed, assessment scores, psychiatric and somatic comorbidities, BMI, smoking and drinking habits, as well as the main findings reported by each study.

**Table 2 diagnostics-15-01683-t002:** Studies comparing proxy biomarker levels between depressive and/or suicidal patients vs. controls.

Study	Marker (Sample)	Subjects	Gender (%F)	Age in Years (SD)	Stage of Mental Disorder/Severity	Scales	Assessment Scores (SD)	Psychiatric Comorbidity	Somatic Comorbidity	BMI (SD)	Smoking	Drinking	Main Findings
Ohlsson et al., 2019 [31]	I-FABP (Plasma) Zonulin (Plasma) sCD14 (Plasma)	MDD = 13 rSA = 54 HC = 17	MDD = 53.8% rSA = 55.5% HC = 47.1%	MDD = 34.5 (11.5) rSA = 38.5 (14.5) HC = 34.4 (11.4)	Acute/Moderate to Severe (Depression); Mild to Severe Risk (Suicidality)	MADRS SUAS	MADRS MDD = 28.7 (7.6) rSA = 21.0 (11.7) HC = 0.8 (1.5) SUAS MDD = 28.3 (6.3) rSA = 38.8 (16.9) HC = 0.8 (2.2)	Only for rSA	Multiple	MDD = 25.9 (8.7) rSA = 25.7 (4.4) HC = 23.1 (3.1)	NR	NR	1. ↓ Zonulin in rSA vs. HC/MDD 2. ↑ I-FABP in rSA vs. HC/MDD 3. = Zonulin, I-FABP, and sCD14 in MDD vs. HC 4. = sCD14 in all groups
Alvarez-Mon et al., 2019 [60]	I-FABP (Serum) Zonulin (Serum) LBP (Serum)	MDD = 22 HC = 14	MDD = 59.1% HC = 57.1%	MDD = 40.3 (8.9) HC = 40.8 (10.5)	Chronic/Mild	HAM-D-17	NR	None	NR	MDD = 26.45 (4.04) HC = 25.26 (3.87)	MDD = 22.7 % HC = 21.4 %	MDD = 13.6 % HC = 7.1 %	1. ↑ I-FABP in MDD vs. HC 2. ↑ LBP in MDD vs. HC 3. = Zonulin in MDD vs. HC
Alvarez-Mon et al., 2021 [61]	I-FABP (Serum) Zonulin (Serum) LBP (Serum)	MDD = 30 HC = 20	MDD = 63.3% HC = 60%	MDD = 43.26 (13.14) HC = 40.4 (12.46)	Chronic/Mild	HAM-D-17	NR	None	NR	MDD = 26.74 (5.41) HC = 25.5 (5.36)	MDD = 33.3% HC = 15%	MDD = 10% HC = 5%	1. ↑ I-FABP in MDD vs. HC 2. ↑LBP in MDD vs. HC 3. = Zonulin in MDD vs. HC
Wu et al., 2023 [62]	I-FABP (Plasma) Zonulin (Plasma) LBP (Plasma)	MDD = 50 HC = 40	MDD = 70% HC = 67.5%	MDD = 16 (range: 14–16) HC = 15 (range: 14–16)	Chronic/Moderate to Severe	HAM-D-17	MDD = 22.5 (range: 19–26) HC = 2 (range: 0–3)	None	None	MDD = 19.78 (range: 17.56–23.44) HC = 20.20 (range: 18.34–23.88)	NR	NR	↑ I-FABP, Zonulin, LBP in MDD vs. HC
Osuna et al., 2024 [63]	I-FABP (Unclear)	MDD = 95 HC = 95	MDD = 58% HC = 58%	MDD = 16.1 (range: 14.9–17.2) HC = 16.0 (range: 14.9–17.1)	Mixed/Moderate to severe	CDRS-R	MDD = 56 (range: 50–62) HC = 18 (range: 17–20)	None	NR	MDD = 0.25 (1.05) (BMI-for-age z-score) HC = 0.16 (1.02) (BMI-for-age z-score)	NR	NR	↑ I-FABP in MDD vs. HC
Chen and Wu [64]	I-FABP (Plasma) sCD14 (Plasma)	nsMDD = 34 sMDD = 44 HC = 42	nsMDD = 64.7% sMDD = 63.6% HC = 54.8%	nsMDD = 43.7 (11.2) sMDD = 41.7 (12.6) HC = 33.6 (12.2)	Chronic/Mild (Depression); Mild to Moderate Risk (Sucidality)	HAM-D BSSI HADS-D PHQ-9	HAM-D nsMDD = 5.8 (4.5) sMDD = 11.5 (5.1) HC = 0.5 (1.2) BSSI nsMDD = 1.3 (3.1) sMDD = 12.5 (7.7) HC = 0.02 (0.2) HADS-D nsMDD = 6.7 (4.9) sMDD = 11.9 (4.5) HC = 2.6 (2.5) PHQ-9 nsMDD = 6.7 (5.6) sMDD = 16.0 (6.0) HC = 2.5 (2.8)	NR (Exclusion criteria = mental retardation, BD, SCZ)	NR	nsMDD = 43.7 (11.2) sMDD = 41.7 (12.6) HC = 33.6 (12.2)	NR	NR	= I-FABP and sCD14 in all groups
Zhong et al., 2022 [65]	I-FABP (Plasma) LBP (Plasma) sCD14 (Plasma)	PSD = 48 HC = 48	PSD = 64.6% HC = 72.9%	PSD = 63 (10) HC = 66 (9.3)	Chronic/Mild	HAM-D-24	PSD = 16.71 (6.56) HC = 4.52 (1.68)	None	PSD Hypertension = 64.6% Diabetes mellitus = 20.8% Pregress stroke = 4.1% HC none	PSD = 26.65 (2.61) HC = 26.26 (2.46)	NR	NR	↑ I-FABP in PSD vs. HC = LBP and sCD14 in PSD vs. HC
Wang et al., 2023 [66]	I-FABP (Plasma)	PSD = 91 nPSD * = 208	PSD = 28.6% nPSD = 29.3	PSD = 65.6 (15.3) nPSD = 63.1 (14.3)	Chronic/Mild	HAM-D-24	PSD = 14.1 (5.5) nPSD = 3.0 (1.7)	None	All participants = ischemic or haemorrhagic stroke PSD Hypertension = 75.2% Diabetes mellitus = 36.3% nPSD Hypertension = 67.8% Diabetes mellitus = 34.6%	NR	PSD = 34.1% nPSD = 34.6%	PSD = 33% nPSD = 43.7%	↑ I-FABP in PSD vs. nPSD
Chojnacki et al., 2022 [21]	Calprotectin (Fecal)	SIBO = 40 HC = 40	SIBO = 57.5% HC = 62.5%	SIBO = 45.2 (9.4) HC = 44.7 (7.3)	Chronic/Mild to Moderate	HAM-D	SIBO = 26.3 (4.26) HC = 9.85 (2.21)	NR	NR	SIBO = 22.4 (2.1) HC = 23.8 (1.6)	NR	NR	↑ Fecal-C in SIBO vs. HC
Papakostas et al., 2013 [67]	A-1-AT (Serum)	MDD = 36 HC = 43	MDD = 36.1% HC = 67.4%	MDD = 42.5 (9.8) HC = 30.0 (8.6)	Chronic/Severe	HAM-D-17	MDD = 21.4 (4.4) HC = NA	NR (Exclusion criteria = BD or SCZ)	NR	MDD = 27.7 (5.8) HC = 24.4 (3.5)	NR	NR	↑ A-1-AT in MDD vs. HC
Brouillet et al., 2023—male [68]	I-FABP (Plasma) LBP (Plasma) LPS (Plasma) Calprotectin (Plasma)	nsBD/MDD = 64 sBD/MDD = 40	nsBD/MDD = 0% sBD/MDD = 0%	nsBD/MDD = 44.4 (13.8) sBD/MDD = 45.1 (14.1)	Acute/Unclear	IDS-C30	nsBD/MDD = 22.7 (14.3) sBD/MDD = 24.9 (13.4)	NR	NR (Exclusion criteria = any acute inflammatory condition)	nsBD/MDD = 25.0 (4.0) sBD/MDD = 27.8 (5.8)	nsBD/MDD = 33.9% sBD/MDD = 46.7%	nsBD/MDD = 30.2% sBD/MDD = 46.2%	= Plasma-C and I-FABP in nsBD/MDD vs. sBD/MDD ↓ LBP in nsBD/MDD vs. sBD/MDD ↑ LPS in nsBD/MDD vs. sBD/MDD
Brouillet et al., 2023—female [68]	I-FABP (Plasma) LBP (Plasma) LPS (Plasma) Calprotectin (Plasma)	nsBD/MDD = 108 sBD/MDD = 93	nsBD/MDD = 100% sBD/MDD = 100%	nsBD/MDD = 40.1 (14.1) sBD/MDD = 40.1 (12.8)	Acute/Unclear	IDS-C30	nsBD/MDD = 24.7 (15.7) sBD/MDD = 31.7 (13.5)	NR	NR (Exclusion criteria = any acute inflammatory condition)	nsBD/MDD = 23.8 (6.3) sBD/MDD = 24.3 (5.3)	nsBD/MDD = 40.4% sBD/MDD = 51.8%	nsBD/MDD = 14.3% sBD/MDD = 30.0%	↓ Plasma-C and LBP in nsBD/MDD vs. sBD/MDD = I-FABP and LPS in nsBD/MDD vs. sBD/MDD
Stewart et al., 2020—HIV+ [69]	sCD14 (Serum)	DAP = 348 nDAP = 1179	DAP = 3% nDAP = 3%	DAP = 51.1 (7.6) nDAP = 51.9 (8.4)	NA	PHQ-9	DAP = 15.9 (5.2) nDAP = 2.7 (2.9)	HIV+; DAP = ↑ hepatitis C infection vs. nDAP	DAP = ↑ cocaine abuse/dependence vs. nDAP	NR	DAP = 61% nDAP = 47%	DAP = 51% nDAP = 40%	↑ sCD14 in DAP vs. nDAP
Stewart et al., 2020—HIV− [69]	sCD14 (Serum)	DAP = 263 nDAP = 574	DAP = 12% nDAP = 9%	DAP = 52.0 (7.9) nDAP = 54.2 (9.8)	NA	PHQ-9	DAP = 16.1 (4.9) nDAP = 2.9 (2.9)	DAP = ↑ hepatitis C infection, diabetes, triglycerides vs. nDAP	DAP = ↑ cocaine abuse/dependence vs. nDAP	NR	DAP = 57% nDAP = 43%	DAP = 49% nDAP = 44%	= sCD14 in DAP vs. nDAP
Just et al., 2024 [70]	LBP (Serum)	AAD = 395 HC = 102	AAD = 78% HC = 70.6%	AAD = 21.4 (2.2) HC = 25.6 (9.1)	Mixed/Mild to Moderate	MADRS-S	AAD = 22.8 (9.3) HC = 6.7 (5.6)	NR	AAD = Multiple comorbidity	AAD = 23.5 (4.9) HC = 22.6 (3.0)	AAD = 25.32% HC = 0.98%	AAD = 10.13%﻿ HC = 4.9%	= LBP in AAD vs. HC
Musil et al., 2011 [71]	sCD14 (Serum)	MDD = 32 HC = 20	MDD = 50% HC = 25%	MDD = 44.6 (11.7) HC = 40.0 (10.4)	Acute/Severe	HAM-D-17	MDD = 24.5 (4.75) HC = NA	None	NR	MDD = 71.1 (15.1) HC = 65.4 (12.3)	MDD = 46.88% HC = 55%	NR	= sCD14 in MDD vs. HC
Zengil and Laloğlu 2022 [72]	Zonulin (Serum) Occludin (Serum)	BD-D = 7 (from BD = 44) HC = 44	BD-D = NR HC = NR	BD-D = NR HC = 41.2 (13.1)	Chronic/Unclear	HAM-D-17	BD-D = NR HC = NR	BD	NR	BD-D = NR HC = 22.6 (3.3)	BD-D = NR HC = 40.9%	BD-D = NR HC = 6.81%	↑ Zonulin in BD-D (and BD) vs. HC ↑ Occludin in BD-D (and BD) vs. HC
Lee et al., 2024 [73]	Zonulin (Serum) I-FABP (Serum)	MD ** = 43 HC = 262	MD = 100% HC = 100%	MD = 46 HC = 46	Unclear	BDI-II	MD = NR HC = NR	NR	MD = 25 PCOS- MD+ = 18, PCOS-MD- = 84	MD = 28.82 (5.75) HC = 27.74 (5.40)	MD = NR HC = NR	MD = NR HC = NR	↓ Zonulin in MD vs. HC ↑ I-FABP in MD vs. HC
Maes et al., 2013 [74]	IgA to LPS (Serum) IgM to LPS (Serum)	MDD = 90 HC = 26	MDD = NR HC = NR	MDD = 43.5 (11.3) HC = 40.3 (11.6)	Acute/Unclear	None	None	MDD = none HC = none	MDD = none HC = none	MDD = NR HC = NR	MDD = NR HC = NR	MDD = NR HC = NR	↑ IgA to LPS in MDD vs. HC ↑ IgM to LPS in MDD vs. HC
Zhou et al., 2018 [75]	LPS (Plasma)	MDD = 14 HC = 14	MDD = 100% HC = 100%	MDD = 29 (range: 23–33) HC = 30 (range: 25–32)	Acute (at first prenatal visit, 8–12 w)/Unclear	EPDS	NR	None	None	MDD = 32.5 (range: 25.5–37.3) HC = 33.0 (range: 26.3–38.3)	MDD = 7.1% HC = 0%	MDD = 0% HC = 0%	↑ LPS in MDD vs. HC
Bai et al., 2021 [76]	A-1-AT (Serum)	nsMDD = 53 sMDD = 20 HC = 86	nsMDD = 60% sMDD = 62.3% HC = 60.5%	nsMDD = 37.34 (13.18) sMDD = 32.34 (14.43) HC = 37.35(13.94)	Acute/Severe (Depression); Mild to Moderate Risk (Sucidality)	HAM-D BSSI	HAM-D nsMDD = 26.80 (7.11) sMDD = 32.34 (8.50) HC = 1.35 (1.48) BSSI nsMDD = 7.55 (1.43) sMDD = 12.64 (2.65) HC = 0.47 (0.88)	None except anxiety disorder	None	nsMDD = 24.61 (1.81) sMDD = 23.41 (2.37) HC = 23.09 (2.14)	nsMDD = 20% sMDD-SI = 30.2% HC = 30.2%	NR	↓ A-1-AT in nsMDD and sMDD vs. HC ↓ A-1-AT in sMDD vs. nsMDD
Bai et al., 2022 [77]	A-1-AT (Serum)	MDD = 56 HC = 56	MDD = 67.8% HC = 64.3%	MDD = 35.11 (16.79) HC = 35.71 (15.99)	Unclear/Severe	HAM-D	MDD = 24.86 (5.94) HC = 0.71 (0.95)	NR	NR	MDD = 20.94 (2.37) HC = 21.23 (4.27)	NR	NR	↓ A-1-AT in MDD vs. HC
Karaoulanis et al., 2014 [78]	A-1-AT (Serum)	MDD = 39 HC = 26	MDD = 100% HC = 100%	MDD = 50.10 (3.95) HC = 48.29 (10.95)	Unclear/Moderate to Severe	HAM-D-17	MDD = 16.85 (5.02) HC = 5.68 (2.52)	MDD = none HC = none	All sample = perimenopausal phase, characterized by the presence of irregular cycles or amenorrhea for less than 12 months	MDD = NR HC = NR	MDD = 36.6% HC = 29.2%	MDD = 0.02% HC = 0	= A-1-AT in MDD vs. HC
Maget et al., 2021 [79]	Zonulin (Serum)	P-Dep *** = 55 P-Eut = 37	P-Dep = 25% P-Eut = 37.8%	P-Dep = P-Eut =	Mixed/Mild to Moderate	HAM-D	P-Dep = 16.25 (4.90) P-Eut = 3.84 (3.24)	P-Dep = BD 81.1% P-Eut = BD 14.3%	NR	P-Dep = 27.81 (5.87) P-Eut = 28.36 (6.57)	P-Dep = 45.3% P-Eut = 42.9%	NR	= Zonulin in P-Dep vs. P-Eut
Dickerson et al., 2017 [80]	IgA to LPS (Plasma)	p-nSA **** = 95 p-pSA = 95 p-rSA = 20 HC = 72	p-nSA = 43% p-pSA = 46% p-rSA = 65% HC = 64%	p-nSA = 35.4 (13.4) p-pSA = 39.3 (13.2) p-rSA = 37.0 (14.7) HC = 33.1 (11.4)	Mixed/Moderate to Severe	BPRS	p-nSA = 48.1 (8.7) p-pSA = 50.0 (7.9) p-rSA = 50.8 (6.4) HC = NA	p-nSA SZ = 39 (41%), BD = 37 (39%), MDD = 19 (20%) p-pSA SZ = 50 (53%), BD = 29 (31%), MDD = 16 (17%) p-rSA SZ = 1 (5%), BD = 6 (30%), MDD = 13 (65%)	NR	p-nSA = 30.7 (8.7) p-pSA = 32.1 (9.5) p-rSA = 29.2 (4.6) HC = 28.1 (7.3)	NR	p-nSA = 13 (14%) p-pSA = 22 (23%) p-rSA = 7 (35%) HC = NA	↑ LPS in p-rSA vs. HC
Maes et al., 2008 [58]	IgM and IgA to LPS (of 6 enterobacteria) (Serum)	MDD = 28 HC = 23	MDD = 60.7 HC = 69.6	MDD = 44.3 (11.2) HC = 40.0 (12.9)	Unclear/Unclear	None	NA	None	Unclear	NR	NR	NR	↑ IgM/IgA to LPS in MDD vs. HC
Maes et al., 2019 [59]	IgM and IgA to LPS (of 6 Gram- bacteria) (Serum)	MD = 22 HC = 96	MD = 46.9 HC = 36.4	MD = 42.4 (13.2) HC = 38.2 (13.3)	Acute/Severe	HAM-D	NR	MD BP1 = 27, BP2 = 25, MDD = 44	Unclear	MD = 25.4 (2.8) HC = 25.3 (3.8)	NR	NR	↑ IgM/IgA to LPS in MD vs. HC
Stevens et al., 2018 [81]	LPS (Plasma) Zonulin (Plasma) I-FABP (Plasma)	DEP/ANX = 22 HC = 28	DEP/ANX = 56% HC = 56%	NR	Unclear/Unclear	None	NA	Unclear	Unclear (but asymptomatic for GI disorders)	NR	NR	NR	↑ LPS, Zonulin, I-FABP in DEP/ANX vs. HC

**Abbreviations: Assessment Tools:** BDI-II = Beck Depression Inventory-Second Edition; BPRS = Brief Psychiatric Rating Scale mild; BSSI = Beck Scale of Suicide Ideation; CDRS-R = Children’s Depression Rating Scale-Revised; EPDS = Edinburgh Postnatal Depression Scale; GAD-7 = Generalized Anxiety Disorder Assessment-7; HADS = Hospital Anxiety and Depression Scale (-A = Anxiety subscore, -D = Depression subscore); HAM-D = Hamilton Depression Scale (-17 = 17-item version, -24 = 24-item version); HSCL-25 = Hopkins Symptom Checklist-25; IDS = Inventory of Depressive Symptomatology (-C30 = clinician-rated 30-item version); MADRS = Montgomery-Åsberg Depression Scale (-S = Self-Rating version); PHQ-9 = Patient Health Questionnaire-9; SUAS = Suicide Assessment Scale. **Disorders:** AAD = Affective and Anxiety Disorders; BD = Bipolar Disorder (-D = Depressive; ns- = without SI; s- = with SI); DAP = Depressed at PHQ-9 (≥10) (n- = not Depressed at PHQ-9); DEP/ANX = Depressive Disorder or Anxiety Disorder; MD = Mood Disorder; MDD = Major Depressive Disorder (ns- = without SI; s- = with SI); PCOS = Polycystic Ovary Syndrome; PSD = Post-Stroke Depression; SCZ = Schizophrenia; SI = Suicidal Ideation; SIBO = Small Intestinal Bacterial Overgrowth. **Others:** A-1-AT = Alpha 1 Antitrypsin; Fecal-C = Fecal Calprotectin; HC = Healthy Controls; I-FABP = Intestinal Fatty-Acid Binding Protein; LBP = Lipopolysaccharide Binding Protein; LPS = Lipopolysaccharide; NA = Not Applicable; NR = Not Reported; SA = recent Suicide Attempt (r- = recent). * nPSD = patients who did not exhibit depressive symptoms after stroke (stroke group); ** no-MD and MD grouping criteria based on BDI-II (no-MD ≤14), GAD-7 (no-MD ≤5), and HSCL-25 (D. ≤1.55, A. ≤1.55), self-report; *** P-Dep = patients with currently mild to moderate depression (HAM-D 10–30); P-Eut = patients currently euthymic (HAM-D < 10); **** p-nSA = patients with no history of any SA; p-pSA = patients with lifetime but not in the previous one-month SA; p-rSA = patients with suicide attempt in the past month. “↓” = significantly lower (*p* < 0.05); “↑” = significantly higher (*p* < 0.05); “=” = no significant difference (*p* > 0.05).

**Table 3 diagnostics-15-01683-t003:** Studies assessing the correlation between depressive symptoms and proxy biomarker levels.

Study	Marker (Sample)	Subjects	Gender (%F)	Age in Years	Stage of Mental Disorder/Severity	Scales	Assesment Scores	Psychiatric Comorbidity	Somatic Comorbidity	BMI	Smoking	Drinking	Correlation Type/Value
Iordache et al., 2022 [82]	I-FABP (Serum) LBP (Serum) Zonulin (Serum) Calprotectin (Fecal)	IBD = 30	50%	Almost 50% in the range 40–60	Unclear	PHQ-9	6.50 [6.00] (median [IQR])	NR	CD = 40%, UC = 60%	NR	26.67%	66.67%	I-FABP ρ = −0.059 (*p* = 0.755) **LBP** ρ = **0.398** (*p* = **0.029**) Zonulin ρ = 0.016 (*p* = 0.934) **Calprotectin ** ρ = **0.416** (*p* = **0.022**)
Liśkiewicz et al., 2021 [83]	I-FABP (Plasma) LBP (Serum) Zonulin (Serum) Calprotectin (Fecal)	MDD = 16	50%	44.0 [34.3–56.3] (median [IQR])	Acute/Moderate to Severe	HAM-D-24	23.0 (21.0–28.5) (median [IQR])	NR	NR	25.0 (22.4–26.7) (median [IQR])	NR	NR	I-FABP ρ = 0 (*p* > 0.05) LBP ρ = 0.2 (*p* > 0.05) Zonulin ρ = 0.15 (*p* > 0.05) Calprotectin ρ = 0.1 (*p* > 0.05)
Rajkovaca Latic et al., 2024 [84]	Calprotectin (Fecal) Beta-defensin 2 (Fecal) sIgA (Fecal) Zonulin (Fecal) Lactoferrin (Fecal)	Parkinson’s disease = 64	25%	66 [57–70] (median [IQR])	Unclear	BDI-II	NR	NR	NR	NR	NR	NR	**Calprotectin ** ρ = **0.255** (*p* = **0.04**) Beta-defensin 2 ρ = 0.117 (*p* = 0.36) **sIgA** ρ = **0.255** (*p* = **0.04**) Zonulin ρ = −0.192 (*p* = 0.13) Lactoferrin ρ = −0.060 (*p* = 0.64)
Brouillet et al., 2023 male [68]	Calprotectin (Plasma) I-FABP (Plasma) LBP (Plasma) LPS (Plasma)	MDD = 104	0%	Unclear (range: 19–79)	Acute/Unclear	IDS-C30	Unclear (range: 0–58)	BD = almost half of participants	NR	Unclear (range: 16.3–41.8)	Unclear	Unclear	Calprotectin ρ = 0.12 (*p* > 0.05) I-FABP ρ = −0.2 (*p* > 0.05) LBP ρ = −0.5 (*p* > 0.05) **LPS** ρ = −**0.22** (*p* < **0.05**)
Brouillet et al., 2023 female [68]	Calprotectin (Plasma) I-FABP (Plasma) LBP (Plasma) LPS (Plasma)	MDD = 201	100%	Unclear (range: 18–70)	Acute/Unclear	IDS-C30	Unclear (range: 0–61)	BD = almost half of participants	NR	Unclear (range: 12.9–53.6)	Unclear	Unclear	Calprotectin ρ = 0.12 (*p* > 0.05) I-FABP ρ = −0.12 (*p* > 0.05) LBP ρ = −0.09 (*p* > 0.05) LPS ρ = −0.08 (*p* > 0.05)
Hussain et al., 2023 [85]	sCD14 (Plasma)	HIV+ = 82 (current depression in 7.4% of the sample)	13.4%	53.2 [9.0] (mean [SD])	Unclear	CES-D	19.4 [8.3] (mean [SD])	Current Substance Abuse (% yes) 7.4%; Lifetime Substance Use Disorderc (% yes) 59.8%	Multiple (mostly metabolic)	NR	NR	NR	sCD14 ρ = 0.03 (*p* = 0.76)
Oktayoglu et al., 2015 [86]	Calprotectin (Serum)	Behçet’s disease = 48	47.9%	35.5 [12.2] (mean [SD])	Unclear	HADS-D	6 [4–8] (median [IQR])	NR	Behçet’s disease	NR	NR	NR	Calprotectin ρ = 0.148 (*p* = 0.315)
Madison et al., 2020 [87]	LBP (Serum) sCD14 (Serum)	Patients at Breast Unit = 305 (with malign = 209 and benign = 106 diagnosis)	100	55.8 [11.1] (mean [SD])	Unclear	CES-D	Unclear (range: 0−49)	NR	Breast Cancer Survivors = 209	28.7 [7.37] (mean [SD])	11.33%	NR	LBP r = 0.111 (p > 0.05) sCD14 r = 0.051 (p > 0.05)
Bellei et al., 2019 [88]	A-1-AT (Plasma)	HS-RLS = 14	71.4%	45.3 [6.7] (mean [SD])	Unclear	BDI	Unclear	NR	NR	NR	57.1%	NR	A-1-AT ρ = 0.053 (p = 0.857)
Hirten et al., 2021 [89]	Calprotectin (Fecal)	UC = 15	60%	33 (median)	Unclear	PSS-4	5 [5–11] (median [IQR])	NR	Multiple	26 (median)	0	NR	Calprotectin PSS-4 not significantly associated with FC
Kiecolt-Glaser et al., 2018 [90]	LBP (Serum) sCD14 (Serum)	Married couples = 43 (total participants = 86)	50%	38.22 [8.18] (mean [SD])	NA	CES-D	Unclear	Past mood disorder = 16; Current mood disorder = 2 (MDD = 1, Dysthymia = 1)	No chronic illnesses	32.07 [5.83] (mean [SD])	0%	NR	Regression model LBP n.s. sCD14 n.s.
Cai et al., 2024 [91]	I-FABP (Serum)	Chronic insomnia disorder = 75	60%	42.00 (27.5–53.0) (median [IQR])	NA	HAM-D	6 [3–7.5] (median [IQR])	None	No severe physical illnesses	21.99 [2.90] (mean [SD])	NR	NR	I-FABP ρ = −0.22 (*p* = 0.057)
Lundgren et al., 2016 [92]	Calprotectin (Fecal)	UC = 55	45%	49 [14.2] (mean [SD])	NA	HADS-D	3.29 [2.69] (mean [SD])	NR	NR	NR	NR	NR	Calprotectin ρ = 0.103 (*p* = 0.317)
Yee et al., 2017 [93]	A-1-AT gene (Whole blood)	FEP = 43	51.2%	29.4 [8.0] (mean [SD])	Unclear	PANSS-D	6.6 [3.5] (mean [SD])	NR	NR	22.8 [5.1] (mean [SD])	25.6%	NR	Regression model Beta = 0.380, *p* = 0.016 (when adjusted for age, gender, smoking status, and chlorpromazine equivalents)
Brys et al., 2020 [94]	LPS (Serum)	Patients on a chronic haemodialysis treatment = 59	36%	61 [15] (mean [SD])	NA	GDS	9.7 [6.3] (mean [SD])	Unclear	NR	25.8 [8.3] (mean [SD])	NR	NR	LPS ρ = 0.21 [−0.04, 0.44], *p* = n.s.
Işık et al., 2020 [95]	Zonulin (Serum)	OCD = 24	45.8%	14.3 [2] (mean [SD])	Unclear	RCADS-CV MDD	10.6 [7.6] (mean [SD])	NR	NR	56.84 [31.37] (mean [SD])—percentile	NR	NR	Zonulin ρ = –0.016 (*p* = 0.942)
Kılıç et al., 2020 [96]	Zonulin (Serum)	BD = 41	56%	36.3 [11.8] (mean [SD])	FR = 21, ME = 20	HAM-D	4.54 [3.20] (mean [SD])	NR	NR	26.4 [2.9] (mean [SD])	44%	NR	Zonulin ρ = NR, *p* = n.s.
Meinitzer et al., 2020 [97]	Zonulin (Serum)	Outpatients admitted to CMA testing (=251; CMA+ = 115, CMA− = 136)	63.7%	40:6 [14.5] (mean [SD])	NA	BDI-II	10 [3.20] (median [IQR])	Unclear	Half of 115 CMA with lactose and/or fructosemalabsorption	NR	NR	NR	Zonulin ρ = 0.012, *p* = 0.854
Varanoske et al., 2022 [98]	LBP (Serum)	Male U.S. Marines = 68	0%	25 [3] (mean [SD])	NA	PHQ-9	0.0 [2.0] (mean [SD])	Unclear	Unclear	26.3 [2.1] (mean [SD])	NR	NR	LBP ρ = −0.374, *p* = **0.013**
Louzada and Ribeiro 2020 [99]	I-FABP (Plasma) LPS (Plasma)	Apparently healthy participants = 74	NR	65–90 (range)	NA	GDS-15	NR	NR	NR	NR	NR	NR	Regression model I-FABP Beta = 0.08, *p* = 0.55 LPS Beta = 0.05, *p* = 0.74

**Abbreviations. Assessment Tools:** BDI = Beck Depression Inventory; BDI-II = Beck Depression Inventory-II; CES-D = Center for Epidemiological Studies Depression Scale; GDS = Geriatric Depression Scale; GDS-15 = Geriatric Depression Scale-15; HADS-D = Hospital Anxiety and Depression Scale-Depression; HAM-D = Hamilton Depression Rating Scale; IDS = Inventory of Depressive Symptomatology (-C30 = clinician-rated 30-item version); PANSS-D = Positive and Negative Syndrome Scale-Depression; PHQ-9 = Patient Health Questionnaire-9; PSS-4 = Perceived Stress Scale; QIDS-SR16 = Quick Inventory of Depressive Symptomatology; RCADS-CV = Revised Child Anxiety and Depression Scales-Child Version; **Biomarkers:** FC = Fecal Calrprotectin; sIgA = secretory IgA; **Disorders:** BD = Bipolar Disorder; CD = Crohn’s Disease; CMA = Carbohydrate Malabsorption; FEP = First Episode Psychosis; HS-RLS = Restless Legs Syndrome at High Severity Grade; IBD = Inflammatory Bowel Disease; MDD = Major Depressive Disorder; ME = Manic Episode; OCD = Obsessive–Compulsive Disorder; UC = Ulcerative Colitis; **Others:** FR = full remission; n.s. = not significant; ρ = Spearman’s Rank Correlation Coefficient; r = Pearson correlation coefficient. Statistically significant biomarkers and corresponding results are shown in bold.

## 6. Differences in Biomarkers in Depressive Patients vs. Controls

Ten studies provided data on circulating levels of I-FABP in patients (n = 448) vs. controls (n = 774) [31,60,61,62,63,64,65,66,73,81]. Seven studies included patients with MDD [31,60,61,62,63,64,73], two studies included patients with Post-Stroke Depression (PSD) [65,66], and one study included patients with a depressive disorder or an anxiety disorder [81]. The pooled estimate showed that I-FABP levels were significantly higher in individuals with depressive symptoms compared to the controls (ES = 0.36; 95% CI = 0.11 to 0.61; *p* = 0.004), with a small ES and high heterogeneity (I^2^ = 71.61%) (Figure 2a). Visual inspection of the funnel plot and Egger’s test (*p* = 0.453) did not suggest publication bias.

Sequentially removing single studies from the analysis did not substantially reduce heterogeneity. After removing the sole outlier study [62], the overall estimate remained significant (ES = 0.28; 95% CI = 0.06 to 0.49; *p* = 0.011; I^2^ = 57.49%).

Eight studies provided data on circulating levels of zonulin in patients (n = 242) vs. controls (n = 462) [31,60,61,62,72,73,79,81]. Five studies included patients with MDD [31,60,61,62,73]. Three studies included more heterogeneous samples, involving i) depressed patients with a diagnosis of bipolar disorder (BD) vs. HCs [72], ii) patients with currently mild to moderate depression vs. currently euthymic individuals [79], and iii) patients with a depressive disorder or an anxiety disorder vs. HCs [81].

The pooled estimate showed that zonulin levels were significantly higher in individuals with depressive symptoms compared to controls (ES = 0.69; 95% CI = 0.02 to 1.36; *p* = 0.044), with a medium ES and evidence of high heterogeneity (I^2^ = 92.12%) (Figure 2b). Sequentially removing individual studies from the analysis did not substantially reduce heterogeneity. Visual inspection of the funnel plot and Egger’s test (*p* = 0.085) did not suggest publication bias.

Five studies provided data on circulating levels of LBP in patients (n = 545) vs. controls (n = 224) [60,61,62,65,70]. Three studies included patients with MDD [60,61,62], while in the comparison with HCs, one study included patients with affective and anxiety disorders [70], and one study included patients with PSD [65]. The pooled estimate showed no significant differences in LBP levels between patients and controls (ES = 0.35; 95% CI = −0.07 to 0.77; *p* = 0.107), with evidence of high heterogeneity (I^2^ = 80.55%) (Figure 2c). Sequentially removing individual studies from the analysis did not reduce heterogeneity. Visual inspection of the funnel plot and Egger’s test (*p* = 0.147) did not suggest publication bias.

Three studies, all from the same research group, provided data on circulating levels of antibodies against bacterial endotoxins in patients (n = 332) vs. controls (n = 120) [59,74,100]. Two studies involved patients with MDD [74,100], and one involved patients with mood disorders (MDs) [59]. The pooled estimate showed a significantly increased level of these antibodies in patients (ES = 0.76; 95% CI = 0.54 to 0.98; *p* < 0.001), with a medium–large ES and no heterogeneity (I^2^ = 0.00%) (Figure 2d). Visual inspection of the funnel plot and Egger’s test (*p* = 0.534) did not suggest publication bias.

Three studies provided data on circulating levels of A-1-AT in MDD patients (n = 95) vs. controls (n = 155) [67,76,78]. The pooled estimate showed no significant differences in A-1-AT levels between patients and controls (ES = 0.39; 95% CI = −0.16 to 0.94; *p* = 0.168), with evidence of high heterogeneity (I^2^ = 75.02%) (Figure 2e). Sequentially removing individual studies from the analysis did not reduce heterogeneity. Visual inspection of the funnel plot and Egger’s test (*p* = 0.514) did not suggest publication bias.

Five studies provided data on circulating levels of sCD14 in patients (n = 738) vs. controls (n = 1880) [31,64,65,69,71]. Three studies included patients with MDD [31,64,71], while two studies presented more heterogeneous samples, involving i) patients with PSD vs. HCs [65], and ii) depressed vs. non-depressed veterans, HIV-positive, or HIV-negative [69].

The pooled estimate showed a significant difference in sCD14 levels between patients and controls (ES = 0.11; 95% CI = 0.01 to 0.21; *p* = 0.038), with a low ES and evidence of low heterogeneity (I^2^ = 10.28%) (Figure 2f). Sequentially removing individual studies from the analysis did not reduce heterogeneity. Visual inspection of the funnel plot and Egger’s test (*p* = 0.238) did not suggest publication bias.

Two studies provided data on circulating levels of LPS in patients (n = 36) vs. controls (n = 42) [75,81]. Since these studies did not meet the inclusion criteria, we have included the meta-analysis of these data in the Appendix A.

## 7. Differences in Biomarkers in Suicidal Patients vs. Controls

Three studies provided data on circulating levels of I-FABP in patients at risk of suicide (n = 231) vs. controls (n = 231) [31,64,68]. The comparisons included i) patients with a recent suicide attempt (rSA) vs. HCs [31], ii) patients with MDD and suicidal ideation vs. HCs [64], and iii) patients with a current major depressive episode during a unipolar or bipolar disorder with vs. without a history of SA [68]. The pooled estimate showed no significant differences in I-FABP levels between patients and controls (ES = 0.24; 95% CI = −0.30 to 0.79; *p* = 0.378), with evidence of high heterogeneity (I^2^ = 86.44%) (Figure 3). Sequentially removing individual studies from the analysis did not reduce heterogeneity. Removing the only study with outlier results [31], the overall estimate remained non-significant and the heterogeneity remained high (ES = −0.02; 95% CI = −0.42 to 0.38; *p* = 0.933; I^2^ = 72.81%). Only two studies were available for zonulin [31,72] and sCD14 [31,64], while LBP and LPS were investigated in only one study, with the sample divided by gender [68] (Appendix A).

## 8. Moderator Analysis for Depressive Symptoms Severity

I-FABP was the only biomarker for which a moderator analysis was possible. Specific data for this biomarker were provided by eight studies: five included patients with mild depressive symptoms (n = 225) vs. controls (n = 332) [60,61,64,65,66], and three included patients with moderate to severe depressive symptoms (n = 158) vs. controls (n = 152) [31,62,63].

The pooled estimate showed i) a trend for higher levels in patients with mild depressive symptoms vs. controls (ES = 0.26; 95% CI = 0.00 to 0.52; *p* = 0.052), with intermediate heterogeneity (I^2^ = 46.67%), and ii) no significant differences for moderate to severe depressive symptoms (ES = 0.58; 95% CI =−0.08 to 1.24; *p* = 0.087), with high heterogeneity (I^2^ = 84.09%) (Figure 4).

Removing the two studies with outlier results [31,66], the results reversed, with the overall estimate being non-significant for patients with mild depressive symptoms (ES = 0.10; 95% CI = −0.14 to 0.35; *p* = 0.404; I^2^ = 0.00%) and significant for patients with moderate to severe depressive symptoms (ES = 0.90; 95% CI = 0.35 to 1.45; *p* = 0.001; I^2^ = 76.77%).

For the other proxy biomarkers, both in relation to the severity of depressive symptoms and suicidality, no further moderator analyses were possible due to the paucity of studies providing useful data. This limitation prevents further investigation into the reasons for heterogeneity in ESs across studies (Appendix A).

## 9. Other Studies Not Included in the Meta-Analysis

One study compared A-1-AT levels in MDD patients vs. HCs, finding a significant decrease in the former [77]. IgA levels to LPS were significantly increased in various patient groups (schizophrenia, BD, and MDD) with a recent (past month) SA vs. HCs [80]. However, no significant difference was found for the other groups (patients with a history of SA in their lifetime or without any SA). Unfortunately, since the data were not provided by the authors, these studies could not be included in our meta-analysis.

## 10. Studies on the Correlation Between Depressive Symptoms and Levels of Proxy Biomarkers

Out of the 19 studies examining the correlation between depressive symptoms and levels of proxy biomarkers, the study populations were highly heterogeneous (Table 3). These studies included patients with mental disorders such as MDD [83], first-episode psychosis [93], obsessive-compulsive disorder [95], BD [96], chronic insomnia disorder [91], HIV+ individuals with current depressive symptoms [85], as well as those with somatic conditions such as inflammatory bowel diseases (IBDs) [82,89,92,101], Parkinson’s disease [84], Behçet’s disease [86], breast cancer [87], restless legs syndrome [88], chronic hemodialysis treatment [94], carbohydrate malabsorption [97], and healthy participants without mental or somatic disorders [90,98,99].

The majority of studies did not find any significant correlation between depressive symptoms and the proxy biomarkers investigated. However, the following positive correlations were observed: i) LBP: a positive correlation was found in one study [82] out of six that investigated it; ii) calprotectin: a positive correlation was observed in two studies [82,84] out of seven that investigated it; and iii) soluble IgA (sIgA): a positive correlation was noted in the only study that investigated it.

Conversely, for LPS, a significantly negative correlation was found in one study out of three, and only for males [68].

## 11. Discussion

In recent years, an increasing number of studies have reported the association between alterations in intestinal permeability (“leaky gut”) and both somatic and mental disorders.

Research has focused on the bidirectional relationship between the GBA and the brain, involving communication through neuroimmune–endocrine mediators [102]. An increasing number of studies are also examining the relationship between gut dysbiosis and mood disorders [103]. Although the pathogenic mechanisms are not yet fully understood, inflammation appears to play a key role [104,105]. It is increasingly evident that stress-induced damage occurs at the level of the BBB and the gut barrier, leading to increased permeability. At the gut level, this phenomenon may be associated with dysbiosis and dysfunction of the GBA, which plays a crucial role in the production of mood-related neurotransmitters, including serotonin [30].

Similarly, IBDs, when untreated or undertreated, may be associated with an increased risk of psychiatric morbidity, particularly depressive and anxiety disorders, as well as suicide [106]. It remains unclear whether mental disorders are a consequence, a precursor, or both in relation to chronic gastrointestinal diseases. Nevertheless, recent studies have demonstrated that IBDs are associated with gut vascular dysregulation, which may facilitate the spread of inflammation to distant sites, including the brain [36]. A possible role appears to be played by the increase in circulating bacterial endotoxins [107].

LPS and other endotoxins can translocate across a compromised intestinal barrier and, once in the bloodstream, trigger the activation of the immune system, particularly through the recognition by Toll-like receptors (TLRs), such as TLR4 for LPS [108]. LBP is a soluble plasma protein that facilitates the transfer of LPS to membrane-bound CD14, which in turn is required to transfer LPS to TLR4 [109]. This activation, as well as other effects indirectly via the pro-inflammatory cytokines TNFα, IL-6, and IL-1β induced by endotoxin, initiates a cascade of pro-inflammatory responses, leading to systemic inflammation and neuroinflammation [110]. Animal studies clearly indicate that endotoxins can contribute to conditions like neurodegenerative diseases [110,111,112].

Preclinical and clinical studies have shown that LPS administration, through various mechanisms other than neuroinflammation—including the regulation of cerebral vascular permeability [36] and tryptophan metabolism [113]—may play a role in the development of mental disorders, particularly depression, anxiety, and sickness behaviors [114,115,116,117]. In the absence of infection, endotoxin still crosses the mucosal membranes of the gut, gums, nose, or lungs, the main source being intestinal permeability [118,119].

Current evidence suggests a correlation between a “leaky gut”, endotoxemia, neuroinflammation, and mental disorders [118], with dysbiosis playing a significant role in compromising the integrity of the intestinal barrier [119].

Recent findings also indicate that biomarkers of gut dysbiosis are increased in patients with mental disorders across diagnostic categories, regardless of medication status. These findings suggest a strong interconnection between mental disorders (particularly depressive disorders) and intestinal dysfunctions, which poses significant challenges in discerning their directionality—that is, the mechanisms through which the mind and body influence each other [45,120]. Nevertheless, growing evidence supports the bidirectional effects of intestinal inflammation and mental disorders [121,122].

Moreover, it appears that even past adverse life experiences may be associated with alterations in the intestinal integrity of the gut barrier, adding to the complexity of the overall picture [123].

The objective of our study was therefore primarily focused on diagnostic purposes, specifically to understand which biomarkers could be utilized by clinicians and forensic pathologists to support their clinical assessment or determination of the manner of death (e.g., natural death or suicide). We found a significant difference between patients with depressive symptoms and controls for biomarkers of intestinal permeability (zonulin), inflammatory response to the passage of bacterial endotoxins (antibodies to endotoxins and sCD14), and acute intestinal epithelial damage (I-FABP).

Regarding I-FABP, our findings differ from those of a previous meta-analysis [45], which included a limited number of studies and incorporated the outlier study by Ohlsson et al [31]. Although no specific methodological flaws were identified in that study, it was characterized by a relatively small sample size (13 MDD patients vs. 17 HCs).

This limited sample size may introduce bias [51,124], potentially accounting for the striking findings reported—namely, that participants with a recent suicide attempt (n = 54) exhibited I-FABP levels not only higher than those of HCs but nearly four-fold greater than those observed in the MDD group.

Through a moderator analysis on I-FABP, we also found moderate reliability of this biomarker in relation to the severity of depressive symptoms. This result is interesting considering that the meta-analysis by Safadi et al. [45] found an increase in the acute phase protein A-1-AT in patients but not in the levels of I-FABP.

This led the authors to consider “sickness behavior” symptoms as being related to systemic inflammation rather than localized to the intestinal tract. Indeed, while I-FABP is specifically expressed in the gut and released upon enterocyte damage [125], serving as a potential marker of intestinal inflammation, A-1-AT is released by various tissues in response to stress, infections, or acute damage, reflecting a broader systemic inflammatory response [126].

Although our study did not identify significant differences in A-1-AT levels, it did reveal evidence suggestive of local inflammatory processes in the gut, consistent with alterations in other biomarkers indicative of intestinal barrier dysfunction, such as zonulin, sCD14 and antibodies against endotoxins.

Regarding suicide risk, our analysis did not identify any specific biomarker that reliably differentiates individuals at risk from the controls. I-FABP was the only biomarker for which sufficient data were available; however, no significant differences were observed. These findings should be interpreted with caution due to the limited number of eligible studies (only three, one of which can be analyzed by gender grouping), the small overall sample size, and methodological heterogeneity. Factors such as a lack of peer review [64], variation in study design [68], and issues related to small sample size [31] may have contributed to inconsistent results. Given these limitations, no definitive conclusions can be drawn about the association between I-FABP—or any other biomarker—and suicide risk.

The limited number of included studies reflects the overall scarcity of research in the literature exploring the relationship between the GBA and mental disorders, despite some recent noteworthy findings [127]. The search for peripheral markers that are easily measurable to support diagnostic–therapeutic pathways in clinical practice and to determine the manner of death in forensic pathology remains imperative [41]. In the psychiatric field, inflammatory markers have demonstrated moderate reliability [40,128,129], although they still find relatively limited application in clinical practice.

In the field of forensic pathology, skepticism persists primarily due to the potential deterioration of blood samples caused by postmortem (PM) changes. Similar challenges have been encountered in other diagnostic contexts (e.g., sepsis, anaphylaxis, and traumatic brain injury). However, some studies provide reassuring findings regarding the reliability of measuring biomarkers in biological fluids collected during autopsy. Examples include procalcitonin, C-reactive protein (CRP), IL-6 [130], tryptase [131], as well as glial fibrillary acidic protein (GFAP), the brain-derived neurotrophic factor (BDNF), and neutrophil gelatinase-associated lipocalin (NGAL) [132].

Although limited, some evidence regarding the PM diagnosis of suicide derives from studies conducted directly on brain tissue rather than peripheral samples. In their meta-analysis, Black and Miller [128] found significantly elevated levels of IL-1β, IL-6, and TNF-α in PM brain samples of suicide victims compared to those of HCs who did not die by suicide. PM levels of IL-4, IL-5, and IL-13 were evaluated in only one study [133], which reported significant increases in IL-4 and IL-13 levels in suicide victims compared to controls. A notable recent study by Liu et al. [134] examined plasma biomarker levels in 30 suicide subjects vs. 25 HCs. Their findings highlighted significant differences in the levels of tryptophan hydroxylase 2 (TPH2), the precursor of BDNF (proBDNF), and the Glial Cell-Derived Neurotrophic Factor (GDNF).

Further evidence is also expected from the study of miRNAs, whose expression has long been found to be altered in suicide completers in specific brain regions [135,136], as well as from research on genetic and epigenetic biomarkers [137].

## 12. Limits

Despite ongoing attention to the relationship between mental and gut health, as evidenced by the addition of new data compared to previous analyses (e.g., Safadi et al. [45]), studies on this topic remain relatively limited. This has led to several limitations: i) a lack of conclusive data regarding the extent of depressive symptoms, primarily due to observed heterogeneity; ii) an inability to draw definitive conclusions about the predictive value of the studied proxy biomarkers for suicidality.

The heterogeneity among the studies has also hindered the investigation of how medications (both somatic and psychiatric), as well as physical illnesses or addictive behaviors, influence the levels of proxy biomarkers.

Recent evidence indicates that psychotropic medications (e.g., antidepressants) may be associated with alterations in biomarkers of intestinal permeability and inflammation, although the results remain conflicting and inconclusive [138,139].

An additional limitation of our findings is the exclusive reliance on cross-sectional study designs, which precludes any inference about temporal or causal relationships, as well as any conclusions regarding the effects of medications on biomarker levels.

The lack of reference normal levels for the biomarkers studied did not allow us to determine whether and to what extent the levels found were below, equal to, or above the norm [137,140]. Finally, further studies should explore additional biomarkers indicative of permeability alterations, such as the claudin family, which are expressed in both the cerebral and intestinal endothelial layers. Altered expression of these biomarkers has been observed in gastrointestinal disorders [141] and certain psychiatric conditions [142]. 

## 13. Conclusions

Our study expands the existing evidence supporting the utility of biomarkers related to intestinal permeability and inflammation in the clinical assessment of depressive symptoms, as well as in the forensic determination of the manner of death. These findings are notably transdiagnostic and, in clinical settings, support the evaluation of patients’ psychopathology and therapeutic processes, allowing a focus on treatment goals. Moreover, they may serve as a potential tool for forensic pathologists, providing additional data indicative of an individual’s mental state at the time of death.

A noteworthy finding is the elevated levels of endotoxins observed among patients with depressive symptoms. This evidence further strengthens the literature on the relationship between dysbiosis, “leaky gut”, endotoxemia, and depressive symptoms, with potential therapeutic implications.

In forensic pathology, the search for biomarkers is particularly valuable, as it remains applicable even in cases with limited anamnestic information. Nevertheless, this field requires further investigation due to the scarcity of studies—particularly those focused on suicide risk and postmortem analyses. Despite these limitations, we consider the findings promising and anticipate that future research will offer deeper insights into the pathophysiological mechanisms underlying the microbiome–GBA.

## Figures and Tables

**Figure 1 diagnostics-15-01683-f001:**
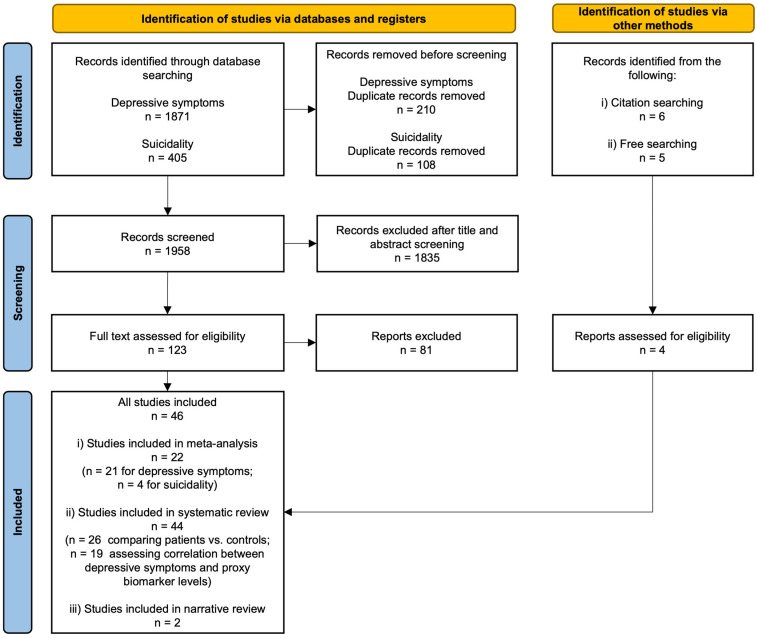
Summary of systematic review performed according to the preferred reporting items for systematic reviews and meta-analysis guidelines.

**Figure 2 diagnostics-15-01683-f002:**
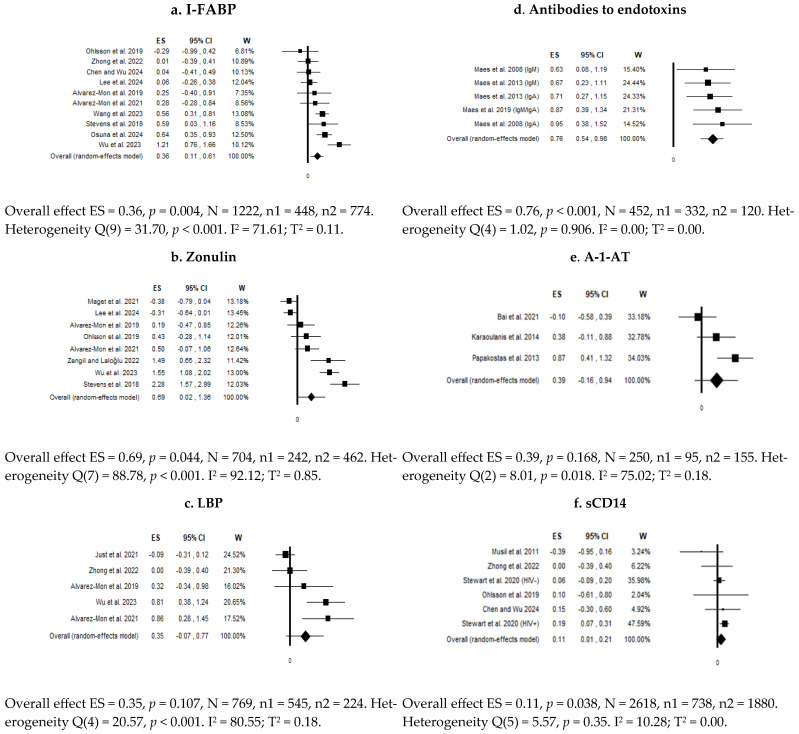
Forest plots of the meta-analysis on overall levels of circulating biomarkers in depressive patients vs. controls. Compared to the null value, values to the left indicate higher levels in controls, and values to the right indicate higher levels in patients.

**Figure 3 diagnostics-15-01683-f003:**
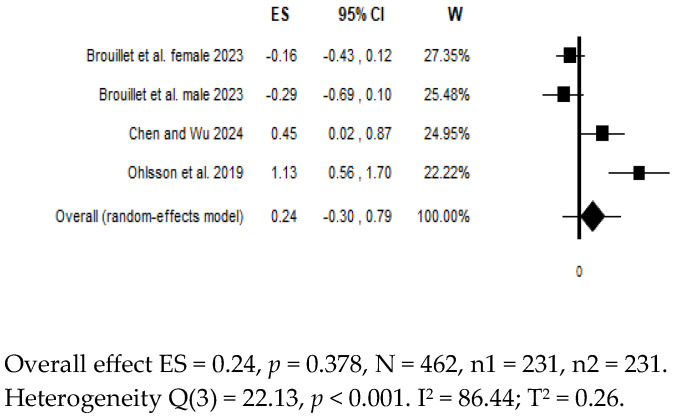
Forest plot of the meta-analysis on overall levels of circulating I-FABP in suicidal patients vs. controls. Compared to the null value, values to the left indicate higher levels in controls, and values to the right indicate higher levels in patients.

**Figure 4 diagnostics-15-01683-f004:**
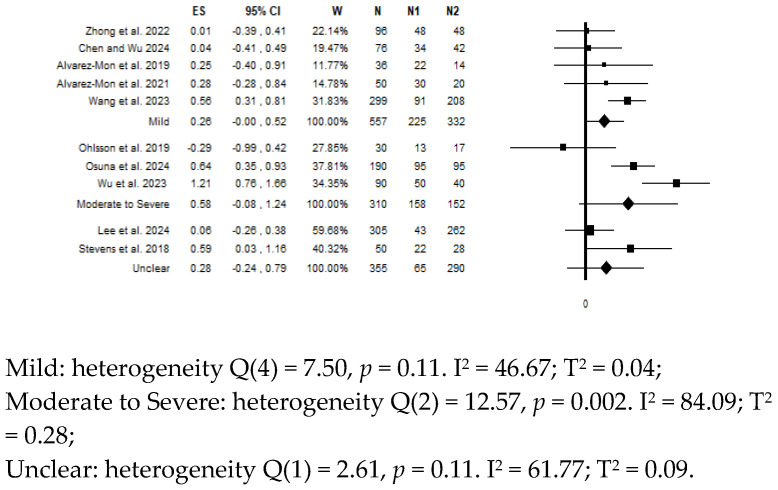
Forest plot of the meta-analysis on levels of circulating I-FABP in depressive patients vs. controls according to a moderator analysis for depressive symptom severity. Compared to the null value, values to the left indicate higher levels in controls, and values to the right indicate higher levels in patients.

**Table 1 diagnostics-15-01683-t001:** Primary biomarkers examined in our study.

Category	Biomarker	Biological Samples *	Description	Variation in Leaky Gut
Structural Proteins	Zonulin	Plasma, serum, feces	A regulatory protein that modulates tight junctions, controlling intestinal permeability.	Increased: Signals dysregulation of tight junctions and barrier compromise.
Structural Proteins	Intestinal fatty-acid binding protein (I-FABP)	Plasma, serum	A cytoplasmic protein released by damaged enterocytes.	Increased: Indicates enterocyte damage due to intestinal barrier disruption.
Immune Proteins	Calprotectin	Plasma, serum, feces	Produced by neutrophils during inflammation. Its increased levels in feces indicate the recruitment of neutrophils to the intestine.	Increased: Indicates inflammation and neutrophil activity due to a compromised barrier.
Immune Proteins	Alpha-1 antitrypsin (A-1-AT)	Plasma, serum	A protease inhibitor that protects tissues and maintains intestinal barrier integrity.	Increased: Indicates protein loss and barrier inflammation in the gut.
Immune Proteins	LPS-binding protein (LBP)	Plasma, serum	Primarily produced by the liver. Acute-phase protein that binds LPS, facilitating immune recognition via CD14 and TLR4.	Increased: Signals systemic immune activation due to bacterial translocation.
Immune Proteins	Soluble CD14 (sCD14)	Plasma, serum	A soluble form of the CD14 protein found in plasma and biological fluids. It regulates the immune response by modulating inflammation triggered by lipopolysaccharides (LPS).	Increased: Indicates systemic immune activation due to endotoxin circulation.
Immune Proteins	Antibodies to endotoxins	Plasma, serum, feces	Immunoglobulins (e.g., IgA and IgM) generated in response to translocated LPS.	Increased: Indicates systemic exposure to bacterial endotoxins due to increased intestinal permeability.
Bacterial Endotoxins	Lipopolysaccharide (LPS)	Plasma	Endotoxin derived from cell envelope of Gram-negative bacteria.	Increased: Indicates bacterial translocation and systemic immune activation.

* refers to the samples used in the studies included in our review.

## Data Availability

All data reported in this paper are shared in the Appendix A.

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
