# Peer review of "Leaky Gut Biomarkers as Predictors of Depression and Suicidal Risk: A Systematic Review and Meta-Analysis"

_diagnostics, 2025, doi:10.3390/diagnostics15131683_

Round 1

Reviewer 1 Report

Comments and Suggestions for Authors

Dear authors, I read your article. The subject is very interesting and I think that the new relation between dysbiosis and depresive symptoms is necessary to be detailed. The number of the patient from the studies, the quantity of data are small, so it is necessary to make more investigations to find the relation between the gut and nervous system. The article can be improved if you give more information about the type of the inflammation of the gut, the type of depression, the dysbiosis,  the treatment. May be the psychiatric treatment determine inflammation of the gut?

Author Response

Dear Reviewer,

We would like to sincerely thank you for your time, availability, and valuable feedback on our manuscript. We have carefully considered your suggestions and revised our work accordingly. All changes made in response to your comments are highlighted in red in the manuscript.

You rightly pointed out the limited sample size, which we have acknowledged in the "Limits" section. This is particularly relevant with regard to data concerning suicidality, which is indeed very scarce.

Regarding your request for more detailed information on the types of depression included, we would like to clarify that these details have been provided in the supplementary material—specifically, in Supplement 1 and, more extensively, in Supplement 4. In Supplement 4, we report for each study the specific type of depression (e.g., AAD = Affective and Anxiety Disorders; BD = Bipolar Disorder; MDD = Major Depressive Disorder; PSD = Post-Stroke Depression, etc.), along with the psychometric scales used. Where available, we have also included data on potential factors influencing inflammatory levels, such as somatic comorbidities, BMI, smoking, and alcohol consumption.

We have explicitly referred to the presence of this information in the Results section of the main text.

Concerning treatment information, unfortunately, most of the included studies only provided general classifications (e.g., antidepressants, antipsychotics) without specifying the exact medications used. As a result, we were unable to extract more detailed data on this aspect. Nevertheless, we have added a note in the "Limits" section to emphasize that existing studies on this topic are limited and inconclusive, although recent, and that future research will likely provide further insights.

Many thanks, sincerely.

Reviewer 2 Report

Comments and Suggestions for Authors

Journal: Diagnostics (ISSN 2075-4418)

Manuscript ID: diagnostics-3645340

Type: Systematic Review

Title: Leaky Gut Biomarkers as Predictors of Depression and Suicidal Risk: A Systematic Review and Meta-Analysis

Authors: Donato Morena * , Matteo Lippi , Matteo Scopetti , Emanuela Turillazzi , Vittorio Fineschi

Section: Clinical Diagnosis and Prognosis

Special Issue: Advances in the Diagnosis and Management of Neuropsychiatric Disorders—2nd Edition

This manuscript presents a timely and well-executed systematic review and meta-analysis of the relationship between leaky gut biomarkers, depression, and suicidality. The study is methodologically sound, PRISMA-compliant, and offers valuable insights into the gut-brain axis with clinical and forensic implications. The use of moderator analysis and a large literature base enhances the work.

The manuscript is overall well-written and informative. Minor revisions are recommended to improve clarity and consistency, particularly addressing minor comments about symptom categorization, figure/table formatting, and abbreviation use. With these changes, the paper will be even more impactful and publishable.

1: The introduction should provide a more specific definition of the hypotheses being tested. The current objective section is broad and slightly repetitive.

2: Can the authors provide a more detailed categorization of depressive symptom severity and suicidality across the included studies, such as specifying the diagnostic tools used, the criteria used, and the time frame of suicide attempts, as these aspects are currently presented ambiguously?"

 3: Converting medians/IQRs to means/SDs was briefly discussed. A more detailed explanation or justification, including an example, would improve transparency.

4: Correct sentences: "This potential source of bias [51, 124] may help explain why, in the same study…" — Suggest restructuring for clarity.

5: Define MDD, LBP, GBA, and other abbreviations in full on the first mention in both the abstract and the body text.

6: Figures 2–4 lack descriptive captions. Include complete titles and concise legends for standalone comprehension.

7: The last sentence of the abstract is ambiguous ("with potential practical applications"). Provide a brief description of potential clinical or forensic applications.

8: Avoid technical phrasing like "no significant differences emerged"; instead, simplify for reader comprehension with "no significant difference was found".

9: Currently, the discussion is mostly about repeating results. Provide a more critical interpretation and contextualize findings in light of recent literature, particularly in the case of conflicting data (for example, differences between I-FABP and A-1-AT).

Author Response

Dear Reviewer,

We sincerely thank you for your availability, expertise, the time you dedicated to reviewing our manuscript, and for your positive feedback. We have carefully considered your suggestions and revised the manuscript accordingly. All changes made in response to your comments are highlighted in blue in the text.

1: The introduction should provide a more specific definition of the hypotheses being tested. The current objective section is broad and slightly repetitive.

We have made an effort to simplify the Introduction and to clarify the objectives of our study.

2: Can the authors provide a more detailed categorization of depressive symptom severity and suicidality across the included studies, such as specifying the diagnostic tools used, the criteria used, and the time frame of suicide attempts, as these aspects are currently presented ambiguously?"

Regarding your request for more detailed information on the types of depression, we have included all relevant details in the supplementary material—specifically in Supplement 1, and more extensively in Supplement 4. In Supplement 4, we report the type of depression for each study (e.g., AAD = Affective and Anxiety Disorders; BD = Bipolar Disorder; MDD = Major Depressive Disorder; PSD = Post-Stroke Depression, etc.), as well as the psychometric scales used.

Where available, we also included information on factors that may influence inflammation levels, such as somatic comorbidities, BMI, smoking, and alcohol consumption. We have indicated the presence of these data in the Results section of the main text.

 3: Converting medians/IQRs to means/SDs was briefly discussed. A more detailed explanation or justification, including an example, would improve transparency.

We have expanded the section and included an example to illustrate the content more clearly.

4: Correct sentences: “This potential source of bias [51, 124] may help explain why, in the same study…” — Suggest restructuring for clarity.

We have made an effort to clarify this section to improve its readability.

5: Define MDD, LBP, GBA, and other abbreviations in full on the first mention in both the abstract and the body text.

We fixed.

6: Figures 2–4 lack descriptive captions. Include complete titles and concise legends for standalone comprehension.

We believe that providing further description beyond the data already presented in the table may be redundant and could overcomplicate the presentation. We thank the reviewer for the suggestion; however, in our opinion, the table already contains all the necessary elements to clearly convey the results.
We also consider it unnecessary to repeat the abbreviations of the markers, as these have already been included in a previous table as well as in the main text.

7: The last sentence of the abstract is ambiguous ("with potential practical applications"). Provide a brief description of potential clinical or forensic applications.

We have revised the Abstract as requested.

8: Avoid technical phrasing like "no significant differences emerged"; instead, simplify for reader comprehension with "no significant difference was found".

We fixed.

9: Currently, the discussion is mostly about repeating results. Provide a more critical interpretation and contextualize findings in light of recent literature, particularly in the case of conflicting data (for example, differences between I-FABP and A-1-AT).

We have made an effort to enrich the Discussion by adding several relevant elements.

Reviewer 3 Report

Comments and Suggestions for Authors

In the present manuscript, authors aim to evaluate the presence of an association, as hypothesised, and the utility of proxy biomarkers in providing clinicians with insights into depressive symptoms and suicidality, independent of specific diagnostic classifications.

Authors said that this investigation may have significant implications in forensic pathology, where biomarkers could serve as valuable tools for determining the cause of death and profiling individuals at risk of suicide.

From my point of view, this review work has relevant and actual information concerning for six biomarkers: I-FABP, zonulin, LBP, antibodies against bacterial endotoxins, A-1-AT, and sCD14 associated with intestinal permeability and inflammation for the clinical assessment of depressive symptoms, as well as for forensic determination of the cause of death, also in different psychiatric pathologies as depression, MDD, and suicide.

The table and figures are understandable; however, they should check the size of letter because there are some very small and it is difficult to read.

I consider the manuscript is very interesting, presentable, clear and understanding.

Also, I observed some minimal errors that should be checked.

1)      In page 1, paragraph 19,

—depressive/suicidal patients vs. healthy controls, or suicidal vs. non-suicidal patients—

The group non-suicidal patients may be, you mean to attempt suicide of patients, resulting in failure suicide. If yes, as a suggestion, you can modify the sentence.

or suicidal vs. attempt suicide of patients

2)      In page 1, paragraph 26,

showed significantly increased levels of I-FABP

Please, as a suggestion, write the name of the acronym I-FABP.

Intestinal fatty acid binding protein (I-FABP).

3)      In page 6, paragraph 189,

or between suicidal vs. non-suicidal patients

Is the same case as point 1

The group non-suicidal patients may be, you mean to attempt suicide of patients, resulting in failure suicide. If yes, as a suggestion, you can modify the sentence.

or suicidal vs. attempt suicide of patients.

Author Response

Dear Reviewer,
We sincerely thank you for your availability, expertise, the time you dedicated to reviewing our manuscript, and for your appreciative comments. We have carefully followed your suggestions in revising our work. All changes made in response to your recommendations are highlighted in purple in the text.

Also, I observed some minimal errors that should be checked.

1)      In page 1, paragraph 19,

—depressive/suicidal patients vs. healthy controls, or suicidal vs. non-suicidal patients—

The group non-suicidal patients may be, you mean to attempt suicide of patients, resulting in failure suicide. If yes, as a suggestion, you can modify the sentence.

or suicidal vs. attempt suicide of patients

In our study, we compared multiple groups based on the data available:

  1. Depressive patients vs. healthy controls (HCs)
  2. Suicidal patients vs. HCs
  3. Suicidal vs. non-suicidal patients

The dimension of suicidality was defined based on recent suicide attempts, current suicidal ideation, or elevated levels of suicidality as measured by standardized assessment tools.
We included this clarification in the Methods section of the main text in order to avoid overloading the Abstract.

2)      In page 1, paragraph 26,

showed significantly increased levels of I-FABP

Please, as a suggestion, write the name of the acronym I-FABP.

Intestinal fatty acid binding protein (I-FABP).

We fixed.

3)      In page 6, paragraph 189,

or between suicidal vs. non-suicidal patients

Is the same case as point 1

The group non-suicidal patients may be, you mean to attempt suicide of patients, resulting in failure suicide. If yes, as a suggestion, you can modify the sentence.

or suicidal vs. attempt suicide of patients.

As mentioned, we have specified the criteria for group selection of patients with elevated suicidality in the Methods section.

Reviewer 4 Report

Comments and Suggestions for Authors

Dear authors,

Thank you for sharing your systematic review and meta-analysis manuscript. The topic you’ve tackled—leaky gut biomarkers as predictors of depression and suicidal risk—is not only scientifically timely but also clinically and forensically significant. The integration of gut-brain axis dysfunction into the neurobiological framework of psychiatric disorders reflects the forward-thinking direction of psychiatric research. Your manuscript demonstrates thoroughness in study selection, rigor in methodological application, and clinical ambition in its implications.

Your work stands at the crossroads of neuroscience, psychiatry, and forensic pathology, an intersection that is gaining increasing relevance. The hypothesis that intestinal permeability may underlie or exacerbate neuropsychiatric symptoms represents a compelling paradigm shift. The attention to detail in your systematic methodology, adherence to PRISMA, and transparent data handling provide a robust foundation for your conclusions. This manuscript not only reinforces known mechanisms but also advances a nuanced biomarker-based approach to depression and suicidality, potentially influencing future diagnostic and preventative strategies.

General Comments (No Bullets)

The manuscript is overall well-written, well-structured, and methodologically sound. It provides an impressive synthesis of the literature and offers a balanced discussion. However, several issues should be addressed to enhance clarity and impact. The introduction, while rich in background, could be more concise to sharpen the focus of the research question. The methods are rigorously applied, but some elements of the statistical handling would benefit from further elaboration or justification—particularly with respect to the handling of high heterogeneity and the decision to pool studies with wide variability in population and biomarker measurement. The results are clear and supported by appropriate figures and supplemental data, though some interpretations in the discussion are slightly overreaching relative to the data. Attention to language flow and grammar will also improve readability.

Section-Specific Comments

Abstract

Lines 12–39 (Page 1)

The abstract is comprehensive but somewhat dense. Consider briefly explaining what I-FABP, zonula, and bacterial endotoxin antibodies are, or at least include one explanatory sentence to orient readers unfamiliar with these biomarkers.

Also, clarify in the conclusion that the data on suicidality were limited in scope—this helps set realistic expectations for the reader.

Introduction

Lines 43–146 (Pages 2–5)

Line 54 (Page 2): Rephrase "close relationship" to a more scientific term like "interconnection" or "bidirectional influence" to better reflect the nuanced interplay.

Lines 87–93 (Page 3): A strong section, but consider rephrasing "alterations...can affect these systems reciprocally" for clarity. Suggested: “Dysregulations within the microbiome-gut-brain axis can exert mutual influences across the involved systems.”

Line 131 (Page 4): Add brief support or citation for the dimensional perspective of depression.

Methods

Lines 147–178 (Pages 5–6)

Line 152 (Page 5): Specify the exact inclusion criteria summarized in Appendix C briefly in the main text to improve readability.

Line 166 (Page 5): The comprehensive nature of the search is a strength, but a short rationale for excluding pre-2000 studies would provide useful context.

Line 171 (Page 5): The PROSPERO registration is a strong point—well done.

Quality of Evidence

Lines 179–186 (Page 6)

Line 185: Acknowledge the potential limitations of using only one type of checklist (CASP for cross-sectional studies), particularly for the included RCT. Consider clarifying how risk of bias in these studies was mitigated or addressed.

Outcomes

Lines 187–227 (Pages 6–7)

Lines 204–205 (Page 7): The preference for Hedges’ g is valid, but explain briefly why it’s preferable to SMD in the context of your dataset (e.g., small sample sizes).

Lines 216–222: The I² interpretation is very well presented and adds clarity to your statistical rigor.

Results

Lines 228–254 (Page 8)

Line 251 (Page 8): Consider briefly stating what constitutes a "small" effect size, as not all readers may recall that 0.36 is considered small.

Line 255: The sentence ends abruptly. Was there a continuation or note regarding sensitivity analysis?

Discussion and Conclusion

Consider clearly separating findings related to depression from those related to suicidality in the discussion.

Avoid overinterpretation of limited data on suicidality. Be cautious about statements that imply causation.

It would be helpful to include a limitations paragraph explicitly addressing heterogeneity, cross-sectional designs, and biomarker variability.

Minor Editorial Suggestions

Consider shortening overly long paragraphs in the introduction for easier reading.

Use consistent terminology for biomarkers across sections (e.g., don’t alternate between “proxy biomarkers” and “biomarkers of intestinal permeability” without clarification).

Several references could be updated or consolidated. I recommend that you add:

Milić J. (2022). How to design a reliable and practical biomarker: the electrophysiologic coefficient of depressiveness - δEPCD. Biomarkers : biochemical indicators of exposure, response, and susceptibility to chemicals, 27(8), 711–714. https://doi.org/10.1080/1354750X.2022.2122565

Milic, J., Jovic, S., & Sapic, R. (2025). Advancing Depression Management Through Biomarker Discovery with a Focus on Genetic and Epigenetic Aspects: A Comprehensive Study on Neurobiological, Neuroendocrine, Metabolic, and Inflammatory Pathways. Genes16(5), 487. https://doi.org/10.3390/genes16050487

Muka, T., Glisic, M., Milic, J., Verhoog, S., Bohlius, J., Bramer, W., Chowdhury, R., & Franco, O. H. (2020). A 24-step guide on how to design, conduct, and successfully publish a systematic review and meta-analysis in medical research. European journal of epidemiology35(1), 49–60. https://doi.org/10.1007/s10654-019-00576-5

Author Response

Dear Reviewer,

We sincerely thank you for your availability, expertise, the time you dedicated to reviewing our manuscript, and for your kind appreciation. We have carefully addressed your suggestions in revising our work. All changes made in response to your comments are highlighted in green in the text.

General Comments (No Bullets)

The manuscript is overall well-written, well-structured, and methodologically sound. It provides an impressive synthesis of the literature and offers a balanced discussion. However, several issues should be addressed to enhance clarity and impact. The introduction, while rich in background, could be more concise to sharpen the focus of the research question.

We have worked to streamline the Introduction and clarify the objectives of the study.

The methods are rigorously applied, but some elements of the statistical handling would benefit from further elaboration or justification—particularly with respect to the handling of high heterogeneity and the decision to pool studies with wide variability in population and biomarker measurement. The results are clear and supported by appropriate figures and supplemental data, though some interpretations in the discussion are slightly overreaching relative to the data. Attention to language flow and grammar will also improve readability.

Section-Specific Comments

Abstract

Lines 12–39 (Page 1)

The abstract is comprehensive but somewhat dense. Consider briefly explaining what I-FABP, zonula, and bacterial endotoxin antibodies are, or at least include one explanatory sentence to orient readers unfamiliar with these biomarkers.

We have added a brief explanation of the most significant biomarkers in the Conclusions section.

Also, clarify in the conclusion that the data on suicidality were limited in scope—this helps set realistic expectations for the reader.

We have added this specification.

Introduction

Lines 43–146 (Pages 2–5)

Line 54 (Page 2): Rephrase "close relationship" to a more scientific term like "interconnection" or "bidirectional influence" to better reflect the nuanced interplay.

We fixed

Lines 87–93 (Page 3): A strong section, but consider rephrasing "alterations...can affect these systems reciprocally" for clarity. Suggested: “Dysregulations within the microbiome-gut-brain axis can exert mutual influences across the involved systems.”

We have rephrased the sentence in an effort to clarify it.

Line 131 (Page 4): Add brief support or citation for the dimensional perspective of depression.

We have expanded this section as per your suggestion.

Methods

Lines 147–178 (Pages 5–6)

Line 152 (Page 5): Specify the exact inclusion criteria summarized in Appendix C briefly in the main text to improve readability.

We have expanded this section as per your suggestion.

Line 166 (Page 5): The comprehensive nature of the search is a strength, but a short rationale for excluding pre-2000 studies would provide useful context.

We have expanded this section (in Appendix C of Supplement 1), as per your suggestion.

Line 171 (Page 5): The PROSPERO registration is a strong point—well done.

Many thanks

Quality of Evidence

Lines 179–186 (Page 6)

Line 185: Acknowledge the potential limitations of using only one type of checklist (CASP for cross-sectional studies), particularly for the included RCT. Consider clarifying how risk of bias in these studies was mitigated or addressed.

We applied the CASP tool also to the only RCT included, namely the study by Osuna et al. [Osuna E, Baumgartner J, Wunderlin O, Emery S, Albermann M, Baumgartner N, Schmeck K, Walitza S, Strumberger M, Hersberger M, Zimmermann MB, Häberling I, Berger G, Herter-Aeberli I; Omega-3 Study Team. Iron status in Swiss adolescents with paediatric major depressive disorder and healthy controls: a matched case-control study. Eur J Nutr. 2024 Apr;63(3):951-963. doi: 10.1007/s00394-023-03313-7.] Since this study, despite being an RCT, was evaluated solely on baseline data, it was effectively treated as a cross-sectional study in our risk of bias assessment. Specifically, the intervention outcomes were not considered, and only baseline data were extracted.

Outcomes

Lines 187–227 (Pages 6–7)

Lines 204–205 (Page 7): The preference for Hedges’ g is valid, but explain briefly why it’s preferable to SMD in the context of your dataset (e.g., small sample sizes).

We have added an explanation for this choice, as suggested.

Lines 216–222: The I² interpretation is very well presented and adds clarity to your statistical rigor.

Many thanks.

Results

Lines 228–254 (Page 8)

Line 251 (Page 8): Consider briefly stating what constitutes a "small" effect size, as not all readers may recall that 0.36 is considered small.

This explanation was already included in the Outcomes section (“Values of 0.20, 0.50, and 0.80 for Hedges’ g were considered indicative of small, medium, and large effects, respectively”).

Line 255: The sentence ends abruptly. Was there a continuation or note regarding sensitivity analysis?

We apologize, but we were unable to locate the sentence to which you refer. In our manuscript, at line 255, we find the following statement:

“Sequentially removing single studies from the analysis did not substantially reduce heterogeneity. Removing the only study with outlier results [65], the overall estimate remained significant (ES=0.28; 95% CI=0.06 to 0.49; p=0.011; I2=57.49%).

Discussion and Conclusion

Consider clearly separating findings related to depression from those related to suicidality in the discussion.

Avoid overinterpretation of limited data on suicidality. Be cautious about statements that imply causation.

We have extensively revised this section, making efforts to incorporate the suggested improvements.

It would be helpful to include a limitations paragraph explicitly addressing heterogeneity, cross-sectional designs, and biomarker variability.

We have added additional notes on the limitations in the dedicated section, also highlighting the potential effects of medications on the investigated biomarkers. However, it was not possible to explore this aspect further due to the limited availability of data.

Minor Editorial Suggestions

Consider shortening overly long paragraphs in the introduction for easier reading.

Use consistent terminology for biomarkers across sections (e.g., don’t alternate between “proxy biomarkers” and “biomarkers of intestinal permeability” without clarification).

We have predominantly used the term “proxy biomarkers,” reserving references to “biomarkers of intestinal permeability” and “inflammatory biomarkers” only when discussing the theoretical framework, rather than the specific data from our study.

Several references could be updated or consolidated. I recommend that you add:

Thank you for the suggestion; we found it useful and have added two references accordingly.

Reviewer 5 Report

Comments and Suggestions for Authors

This peer-reviewed manuscript is devoted to the search of reliable biomarkers of certain neurological diseases based on the gut-brain axis. The authors describe in sufficient detail the methods used in the work, the choice of target references for meta-analysis, and mention an extended list of references for a systematic review.In general, the structure of the article and the quality of the analysis are not in doubt. My comments are technical ones. 

In the main text of the manuscript, in the Methods section, as well as in Figure 1 with the study design, the authors state that 23 articles were included in the meta-analysis and 45 articles were included in the systematic review. Furthermore, in the additional material in the tables and in the descriptions of the articles used, different articles appear under different numbers. It is very difficult to assemble a single picture and verify the correctness of the use of the entire dataset.

For example, “Appendix D-Quality of evidence extended” provides information on 25 publications. Are these publications included in the meta-analysis? Then, which 2 were additionally included in the description, and why were they not included in the analysis?

“Appendix D-Quality of evidence synthesis. Table S2. Quality of evidence of studies included in the quantitative analysis” includes information on 26 publications. Why? Why are there additional articles here?

"Supplement 3 - Included and Excluded Studies", with reasons for Studies identified from the primary search" includes information on 133 publications, but only 46 publications were included in the study taking into account the systematic review (again 1 - additional), while for 34 publications it is indicated that they were included in the study without a comment on why they were not included in the meta-analysis, or the total list of publications for the meta-analysis was 34 publications?

“Supplement 4 - Table S1. Studies comparing depressive and/or suicidal patients vs. controls about proxy biomarkers levels” contains information on data from 28 publications, while “Table S2. Studies assessing the correlation between depressive symptoms and proxy biomarker levels”: only 20 each.

I would like the authors to verify and more clearly describe the data sets for the types of studies they used in their work. Perhaps they included additional information in the study design diagram.

The manuscript has been recommended for publication after addressing the above issues.

Author Response

Dear Reviewer,

We sincerely thank you for your availability, expertise, the time you dedicated to reviewing our manuscript, and for your kind appreciation. We have carefully addressed your suggestions in revising our work. All changes made in response to your comments are highlighted in orange in the text.

In the main text of the manuscript, in the Methods section, as well as in Figure 1 with the study design, the authors state that 23 articles were included in the meta-analysis and 45 articles were included in the systematic review. Furthermore, in the additional material in the tables and in the descriptions of the articles used, different articles appear under different numbers. It is very difficult to assemble a single picture and verify the correctness of the use of the entire dataset.

We sincerely thank the reviewer for their careful and attentive review, which allowed us to identify and correct some embarrassing errors in the counting of studies. We have thoroughly re-examined all numbers, and although the figures may appear complex due to overlaps—for example, some studies providing both quantitative and qualitative data—the final numbers are as follows:

  • The studies included in both the systematic review and meta-analysis are 22 (of which 3 studies—Maes 2013, Stewart, and Brouillet—were split into two separate datasets).
  • Those included only in the systematic review amount to 22, which, combined with the previous group, results in a total of 44 studies.
  • Additionally, 2 studies were included only in the narrative section.

The total number of studies reported in Supplement 4 is therefore 44.

In Supplement 4, we have also distinguished studies including cases versus controls (n = 26) from those reporting correlations (n = 19). Only for 22 of the first 26 studies was it possible to extract data suitable for meta-analysis.

For clarity and in accordance with editorial guidelines, the tables do not show numerical counts but rather provide bibliographic references for each study.

We understand that evaluating the methodology may be challenging under these circumstances; nevertheless, we have made every effort to conduct this assessment as accurately as possible.

For example, “Appendix D-Quality of evidence extended” provides information on 25 publications. Are these publications included in the meta-analysis? Then, which 2 were additionally included in the description, and why were they not included in the analysis?

“Appendix D-Quality of evidence synthesis. Table S2. Quality of evidence of studies included in the quantitative analysis” includes information on 26 publications. Why? Why are there additional articles here?

We have revised the qualitative analysis conducted. The number of studies examined (i.e., those included in the quantitative analysis) is now correctly reported as 22.

"Supplement 3 - Included and Excluded Studies", with reasons for Studies identified from the primary search" includes information on 133 publications, but only 46 publications were included in the study taking into account the systematic review (again 1 - additional), while for 34 publications it is indicated that they were included in the study without a comment on why they were not included in the meta-analysis, or the total list of publications for the meta-analysis was 34 publications?

We have thoroughly revised the list, specifying which studies were included in the quantitative analysis as well as in the systematic review, and which were included only in the systematic review or the narrative review. Once again, the numbers correspond to those previously reported. The PRISMA flowchart has also been updated accordingly in the manuscript.

“Supplement 4 - Table S1. Studies comparing depressive and/or suicidal patients vs. controls about proxy biomarkers levels” contains information on data from 28 publications, while “Table S2. Studies assessing the correlation between depressive symptoms and proxy biomarker levels”: only 20 each.

The two tables have been revised and now include 26 studies (Table S1) and 19 studies (Table S2). It should be noted that Brouillet et al. 2023 provide both group comparison data and correlation data, so the correct total number of studies remains 44.

I would like the authors to verify and more clearly describe the data sets for the types of studies they used in their work. Perhaps they included additional information in the study design diagram.

We have corrected the PRISMA flowchart.

The manuscript has been recommended for publication after addressing the above issues.

Many thanks again, sincerely.

Round 2

Reviewer 1 Report

Comments and Suggestions for Authors

Dear authors, I like your opinions, I see all the modifications and now the article is complete.